# Ground solar absorption observations of total column CO, CO$_2$, CH$_4$, and aerosol optical depth from California's Sequoia Lightning Complex Fire: emission factors and modified combustion efficiency at regional scales

**Isis Frausto-Vicencio[1], Sajjan Heerah[2], Aaron G. Meyer[2], Harrison A. Parker[3], Manvendra Dubey[2], and Francesca M. Hopkins[1]**

[1]Department of Environmental Sciences, University of California, Riverside, CA 92521, USA
[2]Earth and Environmental Sciences Division, Los Alamos National Laboratory, Los Alamos, NM 87545, USA
[3]Division of Geological and Planetary Science, California Institute of Technology, Pasadena, CA 91125, USA

**Correspondence:** Isis Frausto-Vicencio (ifrau001@ucr.edu)

**Abstract.** CE1 With global wildfires becoming more widespread and severe, tracking their emissions of greenhouse gases and air pollutants is becoming increasingly important. Wildfire emissions have primarily been characterized by in situ laboratory and field observations at fine scales. While this approach captures the mechanisms relating emissions to combustion phase and fuel properties, their evaluation on regional-scale plumes has been limited. In this study, we report remote observations of total column trace gases and aerosols during the 2020 wildfire season from smoke plumes in the Sierra Nevada of California with an EM27/SUN solar Fourier transform infrared (FTIR) spectrometer. We derive total column aerosol optical depth (AOD), emission factors (EFs) and modified combustion efficiency (MCE) for these fires and evaluate relationships between them, based on combustion phase at regional scales. We demonstrate that the EM27/SUN effectively detects changes in CO, CO$_2$, and CH$_4$ in the atmospheric column at $\sim 10$ km horizontal scales that are attributed to wildfire emissions. These observations are used to derive total column EF$_{CO}$ of $120.5 \pm 12.2$ and EF$_{CH_4}$ of $4.3 \pm 0.8$ for a regional smoke plume event in mixed combustion phases. These values are consistent with in situ relationships measured in similar temperate coniferous forest wildfires. FTIR-derived AOD was compared to a nearby AERONET (AErosol RObotic NETwork) station and observed ratios of AOD to X$_{CO}$ were consistent with those previously observed from satellites. We also show that co-located X$_{CO}$ observations from the TROPOspheric Monitoring Instrument (TROPOMI) satellite-based instrument are $9.7 \pm 1.3$ % higher than our EM27/SUN observations during the wildfire period. Finally, we put wildfire CH$_4$ emissions in context of the California state CH$_4$ budget and estimate that $213.7 \pm 49.8$ Gg CH$_4$ were emitted by large wildfires in California during 2020, about 13.7 % of the total state CH$_4$ emissions in 2020. Our work demonstrates a novel application of the ground-based EM27/SUN solar spectrometers in wildfire monitoring by integrating regional-scale measurements of trace gases and aerosols from smoke plumes.

# 1 Introduction

Wildfires have become deadlier, more destructive, and more frequent globally over the past few years (UNEP, 2022). Particularly, the 2020 wildfire activity season saw massive wildfires in the western USA, Australia, Brazil, and the Arctic. The California 2020 wildfire season was exacerbated by abnormally high temperatures and dry conditions (Jain et al., 2022; Cho et al., 2022) and emitted 10 times more carbon dioxide into the atmosphere than the 2000–2019 annual average wildfire emissions (CARB, 2020). In the San Joaquin Valley (SJV) of California, atmospheric concentrations of fine air pollutant particles that are 2.5 μm or smaller in size, also known as particulate matter 2.5 ($PM_{2.5}$), were found to be 4 times higher during the 2020 fire season than non-fire periods (Ahangar et al., 2022). The high temperatures and dry conditions, combined with moisture from a tropical storm, led to a dry lightning storm event in August 2020, where lightning-ignited wildfires burned more acres in California than at any other time in recorded history (Morris and Dennis, 2020). This included the lightning-sparked Castle Fire (part of the Sequoia Lightning Fire (SQF) Complex) that killed 10 %–14 % of the large sequoias in the Sierra Nevada and has become the largest fire in a giant sequoia grove on record (Stephensen and Brigham, 2021). Historic fire suppression and land use changes in this area have led to an increase in wildfires burning at higher intensity and larger areas (Moody et al., 2006; Scholl and Taylor, 2010). Climate change has increased the forest fire activity in the western USA (Zhuang et al., 2021) and will increase the likelihood of wildfires in the Sierra Nevada, with greater burned area due to higher daily temperatures (Gutierrez et al., 2021) and implications for air quality and carbon emissions (Navarro et al., 2016).

Wildfires are a major source of air pollutants, including particulate matter (PM), carbon monoxide (CO), and greenhouse gases, primarily carbon dioxide ($CO_2$) and methane ($CH_4$; Akagi et al., 2011; Wiedinmyer et al., 2011; Andreae, 2019). The high levels of PM and CO released from fires are dangerous to human health and degrade air quality on a local, regional, and global scale (Schneising et al., 2020; Aguilera et al., 2021). CO is an air toxic and is considered an indirect greenhouse gas, as it is a major sink for the hydroxyl radical (OH), increasing the abundance of $CH_4$ through photochemical feedbacks (Li et al., 2018), and also produces ozone ($O_3$), a short-lived greenhouse gas. $CO_2$ and $CH_4$ are the dominant greenhouse gases and are responsible for most of the current anthropogenic climate change (IPCC, 2021). Although emissions from fires are biogenic sources of $CO_2$, they are released rapidly compared to the slow timescales of carbon uptake required to grow vegetation fuels. Increased fire activity increases atmospheric $CO_2$ in the short term and can locally alter the terrestrial carbon cycle balance by reducing photosynthetic $CO_2$ uptake due to high levels of vegetation disturbance (Lasslop et al., 2019). While $CO_2$ losses can be estimated as a function of burned area and fuel consumption, emissions of CO, $CH_4$, and aerosols are more difficult to estimate because they vary greatly with wildfire combustion phases. As global wildfires become more widespread and severe, tracking emissions of greenhouse gases and air pollutants from smoke will become increasingly important for efforts to track emissions of greenhouse gases and understand the impacts of fire on the atmosphere (Aguilera et al., 2021; Wilmot et al., 2022).

Our understanding of the atmospheric impacts of increasing fire activity relies on accurate observations and a process-based estimation of fire emissions that have been developed using in situ measurements (Urbanski, 2014). While several space-based instruments can retrieve and derive emissions of important trace gases globally, observations of trace gases are limited by spatiotemporal coverage and aerosol burden from smoke plumes (Schneising et al., 2020). Recent satellite studies have focused on trace gas emissions and ratios for $CH_4$, CO, nitrogen oxides ($NO_x$), and ammonia ($NH_3$; Whitburn et al., 2015; Adams et al., 2019; Griffin et al., 2021; Jin et al., 2021), but few focus on the integration of trace gases and aerosols. Ground-based solar spectrometers present an alternative technique to measure and understand fire emissions at regional scales and temporally complement satellite observations. Column measurements are insensitive to the planetary boundary layer growth and are less affected by nearby point sources than in situ measurements, making them a good candidate for regional-scale monitoring (Lindenmaier et al., 2014). The EM27/SUN is a ground-based remote sensing instrument that is relatively portable and robust for field deployments (Chen et al., 2016; Heerah et al., 2021). These instruments are the basis for the ground-based network of the Fourier transform infrared (FTIR) COCCON (COllaborative Carbon Column Observing Network; Frey et al., 2019; Vogel et al., 2019; Alberti et al., 2022a, b), which complements the NDACC (Network for the Detection of Atmospheric Composition Change; Bader et al., 2017; De Mazière et al., 2018) and TCCON (Total Column Carbon Observing Network), two high-resolution FTIR trace gas monitoring networks (Toon et al., 2009; Wunch et al., 2011).

Field-based measurements of biomass burning in temperate forests are limited and sparse (Burling et al., 2011; Urbanski, 2014), despite the increase in burning activity in the western USA (Zhuang et al., 2021). The EM27/SUN provides vertically integrated column measurements of $CH_4$, $CO_2$, and CO, which allows for the calculation of the modified combustion efficiency (MCE) and emission factors (EFs) in the total column of smoke plumes downwind of wildfires. MCE values give insight into the relative amounts of flaming and smoldering combustion of the fire. EFs are defined as the mass of gas or aerosol emitted per dry biomass consumed and are critical inputs for models to accurately calculate emissions and construct wildfire inventories (Urbanski, 2014). Providing new EFs will help improve regional biomass burning estimates. Past studies have derived atmo-

spheric column-based EFs with respect to CO from wildfires, using solar FTIR spectrometers (Paton-Walsh et al., 2005; Viatte et al., 2014, 2015; Lutsch et al., 2016, 2020; Kille et al., 2022). The observed small changes in $CO_2$ with respect to the large atmospheric background has limited previous FTIR-based studies in their ability to derive EFs with respect to $CO_2$. This has consequently inhibited the calculation of MCE. Here, we present the first EFs with respect to $CO_2$ and MCE for wildfires calculated by total column FTIR.

During part of the 2020 wildfire season, we deployed an EM27/SUN in the SJV downwind of two major Sierra Nevada wildfires, the SQF Complex (which comprised the Castle and Shotgun fires) and the Creek Fire. We report $EF_{CO}$ and $EF_{CH_4}$ from the SQF Complex, a mixed conifer forest wildfire, and calculate the wildfire's combustion phase with MCE values. We also derived the aerosol optical depth (AOD) from the EM27/SUN solar spectra and compare to a nearby AERONET (AErosol RObotic NETwork) site. Furthermore, because ground-based column measurements operate on similar scales as satellites (McKain et al., 2015), we compared EM27/SUN measurements with observations of CO from TROPOspheric Monitoring Instrument (TROPOMI) collected during the fires. Finally, using enhancement ratios, we estimate wildfire $CH_4$ emissions for 2020 and put our 2020 wildfire $CH_4$ emission estimates in context of the California state $CH_4$ budget.

## 2 Data sources and methods

We measured the column-averaged dry air mole fractions ($X_{gas}$) of $CH_4$, $CO_2$ and CO ($X_{CH_4}$, $X_{CO_2}$, and $X_{CO}$) with the EM27/SUN at a site downwind of two major fires in the Sierra Nevada, namely the SQF Complex and the Creek Fire. We also derived AOD from the measured solar spectra of the EM27/SUN and compare to a nearby AERONET site (Fig. 1). The measurement site was located 60 km west of the SQF Complex that was composed of the Castle and Shotgun fires and 80 km south of the Creek wildfire (Fig. 1a). The SQF Complex fires began on 19 August 2020, after a dry thunderstorm and lightning event ignited the fires in the Sierra Nevada. By 12 September, the SQF Complex had grown to 283 $km^2$, and a large wildfire plume from this fire traveled west, directly over our measurement site, that was captured by the EM27/SUN and TROPOMI (Fig. 1b and c). The Creek Fire began on the evening of 5 September and high upper-level winds produced a pyrocumulus cloud on 6 September that reached an altitude over 15 km (Morris and Dennis, 2020). Smoke filled the valley, and smoky, overcast skies remained over large parts of the SJV for the next 2 weeks as fires kept burning. In total, the SQF Complex burned 686 $km^2$ and Creek burned 1515 $km^2$, placing both these fires among the 20 largest California wildfires ever recorded (Morris and Dennis, 2020).

## 2.1 EM27/SUN atmospheric column observations

The Bruker Optics EM27/SUN solar-viewing Fourier transform spectrometer, owned by Los Alamos National Laboratory (LANL), collected continuous daytime column measurements in Farmersville, California (36.31, −119.19), from 8 September until 17 October 2020, for a total of 40 d of observations. The EM27/SUN $X_{gas}$ values were retrieved from unaveraged, double-sided interferograms using the interferograms to spectra (I2S) and GFIT (GGG2014 version; https://tccon-wiki.caltech.edu/, last access: 15 September 2022) retrieval algorithms automated by the EM27/SUN GGG interferogram (EGI) processing suite (Hedelius et al., 2016). Surface pressure is required to retrieve dry air columns in GGG, and we used a Coastal Environmental Systems ZENO Weather Station to record surface pressure at our field site for retrievals. Retrievals also require atmospheric profiles of temperature, pressure, altitude, and water, and these profiles were extracted from the NCEP/NCAR (National Centers for Environmental Prediction/National Center for Atmospheric Research) reanalysis product (Kalnay et al., 1996). We calibrated the EM27/SUN via co-located measurements alongside the IFS125, a high spectral resolution FTIR operated by TCCON at the California Institute of Technology (CIT), both before and after the collection periods, to determine calibration factors ($R_{gas}$), assuming a linear model forced through the origin for each gas (e.g., $X_{TCCON} = X_{EM27} R_{gas}$; Chen et al., 2016; Hedelius et al., 2016). The TCCON network sets the standard as the current state-of-the-art, ground-based validation system for remote sensing and satellite-based observations of greenhouse gases (Wunch et al., 2011), and TCCON observations are tied to the World Meteorological Organization (WMO) standard greenhouse gas scale. Co-locating the EM27/SUN and TCCON instruments ensures system stability of the EM27/SUN after transportation to field sites. Co-located measurements were performed on 2–3 September 2020 and 30 October– 1 November 2020. Results of the correction factors from the co-located measurements are shown in Table A1. The TCCON instrument also uses the GFIT retrieval algorithm with the same a priori profiles; however, due to different instrument spectral resolutions and averaging kernels, we correct for the differences between the EM27/SUN and TCCON instrument, following Hedelius et al. (2016; Eq. A4), to adjust the EM27/SUN retrievals before comparing with TCCON and deriving calibration factors.

The EM27/SUN solar spectrometer has been previously used to study emissions from urban and agriculture $CH_4$ and $CO_2$ sources (Chen et al., 2016; Viatte et al., 2017; Dietrich et al., 2021; Heerah et al., 2021; Makarova et al., 2021; Alberti et al., 2022a). The recent addition of a CO detector in Bruker's EM27/SUN FTIR spectrometer increases the instrument's utility for measuring combustion sources and as a validation tool for TROPOMI column $X_{CO}$, as it covers the same spectral region (Hase et al., 2016). The EM27/SUN

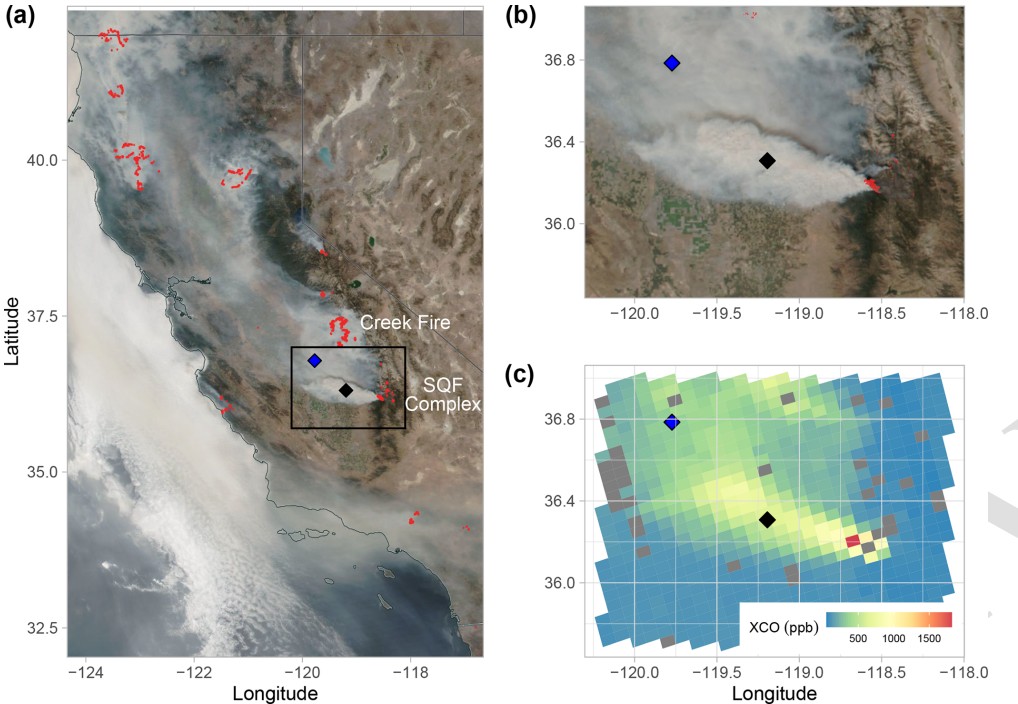

**Figure 1. (a)** Satellite imagery, captured by NOAA-20 Visible Infrared Imaging Radiometer Suite (VIIRS), of heavy smoke in California on 12 September 2020, highlighting fire and thermal anomalies in red (NASA Worldview; https://worldview.earthdata.nasa.gov/, last access: 15 July 2022), with a black diamond shape showing the EM27/SUN measurement location and a blue diamond shape showing the AERONET observational site. **(b)** The inset shows more detail of the smoke plume within the SJV from the SQF Complex in the Sierra Nevada. **(c)** The inset of the TROPOMI $X_{CO}$ overpass on 12 September 2020 at 13:54 PDT.

uses the Sun as the light source which allows it to derive AOD, as demonstrated by Barreto et al. (2020) at the TCCON FTIR and AERONET site ath the Izaña Atmospheric Observatory, Spain. In their study, TCCON spectra were degraded to the same resolution as the EM27/SUN ($0.5 \, \mathrm{cm}^{-1}$), and it was concluded that EM27/SUN spectra would be able to effectively derive AOD. Following their approach, we derive AOD for the wildfire period from our measurements. Further details of the AOD calculation are found in Appendix B.

Prior to measurements in California, the EM27/SUN was stationed in Fairbanks, Alaska, for several months. Given the different settings used with the CAMTRACKER, the solar disk was not centered on the camera, and this misalignment was found on 7 September. Based on co-located measurements with the CIT TCCON on 2 and 3 September, it was determined that the observations within the second detector of $X_{CO}$ were affected on the days prior when the camera was misaligned (2, 3, 6, and 7 September). For this reason, we report measurements of $X_{CO}$, $X_{CO_2}$, and $X_{CH_4}$ beginning on 8 September and use the 30 October–1 November co-located measurements to calculate correction factors for all gases. AOD was derived from micro-windows within the first detector; thus, calculations of AOD were not affected.

## 2.2    TROPOMI CO column measurements

TROPOMI is an instrument launched in late 2017 on board the European Space Agency's (ESA) Sentinel-5 Precursor (S5P). The instrument measures Earth radiance spectra in the ultraviolet, near-infrared, and shortwave infrared, allowing for measurements of a wide range of atmospheric trace gases and aerosol properties (Veefkind et al., 2012). The satellite has a sun-synchronous orbit, with daily global coverage and a spatial resolution of $5.5 \times 7 \, \mathrm{km}^2$ for $CH_4$ and CO operational level 2 (L2) products. The offline (OFFL) CO total column L2 data product filtered for quality assurance values $> 0.5$ are used in this work, as recommended in the product readme file (https://sentinel.esa.int/documents/247904/3541451/ Sentinel-5P-Carbon-Monoxide-Level-2-Product-Readme-File, last access: 4 August 2022). This selection filters out high solar zenith angles, any corrupted retrievals, and influences from high clouds. The majority of the TROPOMI $X_{CH_4}$ product was flagged out near the observational site during our measurement period and hence was not included in this analysis. Following Sha et al. (2021), the TROPOMI CO column densities were converted to $X_{CO}$ (parts per billion – ppb) by using the modeled surface pressure and total column of water to calculate the column of dry air.

There is growing interest in using the TROPOMI $X_{CO}$ product for understanding global wildfire fluxes; however, few studies focus on evaluating those observations (e.g., Jacobs, 2021; Rowe et al., 2021). We measured a range of $X_{CO}$ levels of mixed smoke plumes with our EM27/SUN and were able to isolate a concentrated smoke plume from a nearby fire. This allowed for a ground-based evaluation of the TROPOMI sensor under various wildfire conditions, including high $X_{CO}$ and aerosol loading in the atmosphere. A correction factor was calculated for the EM27/SUN to account for differences in the a priori profile used in the retrieval of $X_{CO}$ in both instruments. We follow the a priori substitution method described in Jacobs (2021) and Sha et al. (2021) to calculate an additive factor for the EM27/SUN. Due to the possibility of measuring narrow smoke plumes on subgrid spatiotemporal scales, we perform a sensitivity study to determine the best co-location criteria for the EM27/SUN to TROPOMI comparison by varying the maximum radius (5–50 km) from the observational site and averaging time (5–30 min) for the EM27/SUN measurements around the TROPOMI overpass time. We required a minimum threshold of at least three 1 min averages of EM27/SUN retrievals within the averaging time aggregations.

## 2.3 AERONET data

AERONET (https://aeronet.gsfc.nasa.gov/, last access: 15 June 2022) is a global network of Sun/sky radiometers, with over 600 sites operated around the globe. AERONET observations include measurements of AOD, microphysical, and radiative properties. The stations are frequently calibrated, and they set the standard for aerosol measurements and validation for satellite products (Giles et al., 2019). AERONET measures AOD at several spectral windows from 340, 380, 440, 500, 675, 870, 940, 1020, and 1640 nm. The Ångström exponent (AE), describing the wavelength dependence of aerosol optical thickness, is calculated from the spectral AOD. We used the AERONET Level 2.0 Version 3 AOD and AE data from the Fresno_2 site (36.78; −119.77) that has been operating in the same location since 2012. This site is located about 90 km away from our EM27/SUN measurement site. Further quality control information can be found in Giles et al. (2019).

## 2.4 Estimating emission factors and modified combustion efficiency

We demonstrate the capability of ground-based solar column measurements to calculate important variables for fire research, including EFs and MCE, for determining fire emissions and understanding different combustion phases of wildfires. As a case study, 12 September observations were selected, as this day had the highest observed $X_{CO}$ and a dominant influence from the SQF Complex (Fig. 1b). We estimate emission ratios of $CH_4$ and CO ($ER_i$) by calculating the slope

from a York linear regression of CO and $CH_4$ excess mole fractions ($\Delta X_i$) relative to excess mole fraction of ($\Delta CO_2$), as shown in Eq. (1). The York linear regression considers the instrument errors along both axes.

$$\text{ER}_i = \frac{\Delta X_i}{\Delta X_{CO_2}} = \frac{X_{i,\text{Fire}} - X_{i,\text{Bkgd}}}{X_{CO_2,\text{Fire}} - X_{CO_2,\text{Bkgd}}} \tag{1}$$

Emission factors of $CH_4$ and CO ($EF_i$) were then calculated, as shown in Eq. (2), by multiplying the $ER_i$ by the molar mass of either CO or $CH_4$ ($MM_i$), divided by the molar mass of carbon ($MM_C$) and total carbon emitted ($C_T$), while assuming $500 \pm 50$ g of carbon is emitted per kilogram of dry biomass consumed ($M_{\text{Biomass}}$; Burling et al., 2010; Akagi et al., 2011). $C_T$ is given by Eq. (3), where $n$ is the number of carbon-containing species measured, $N_j$ is the number of carbon atoms in species $j$, and $\Delta X_j$ is the excess mixing ratio of species $j$ (Yokelson et al., 1999).

$$\text{EF}_i = \frac{\text{ER}_i}{C_T} \cdot \frac{\text{MM}_i}{\text{MM}_C} \cdot M_{\text{Biomass}} \tag{2}$$

$$C_T = \sum_{j=1}^{n} N_j \times \frac{\Delta X_j}{\Delta X_{CO_2}} \tag{3}$$

The MCE is commonly used as a relative measure between the smoldering and flaming combustion phases. Smoldering emissions have an MCE from 0.65–0.85, pure flaming emissions have an MCE of 0.99, and emissions near 0.9 have roughly equal amounts of flaming and smoldering combustion (Akagi et al., 2011). MCE was calculated by dividing the excess mole fraction of $CO_2$ by the sum of the excess mole fractions of $\Delta CO$ and $\Delta CO_2$, as follows:

$$\text{MCE} = \frac{\Delta X_{CO_2}}{\Delta X_{CO} + \Delta X_{CO_2}}. \tag{4}$$

Due the difference in averaging kernels across the trace gases, an averaging kernel correction is applied to Eqs. (1) and (4) (see Appendix C). The enhancement over background mixing ratios ($\Delta X_i$) for each measurement day was calculated by subtracting the background ($X_{i,\text{Bkgd}}$) determined as the 2nd percentile of the daily measured mixing ratios ($X_i$). A sensitivity test showed that emission ratios did not change significantly if the background was calculated using the 1st–5th percentiles. Leveraging the comparison between our ground-based instrument and TROPOMI, we compared the spatial background to show that the 2nd percentile was appropriate (Fig. S2 in the Supplement). The monthly background in September was 411.3 ppm (parts per million) for $X_{CO_2}$, 99.4 ppb for $X_{CO}$, and 1905.3 ppb for $X_{CH_4}$. The monthly average mixing ratios measured in situ at Mauna Loa for $CO_2$ were $411.5 \pm 0.2$ ppm and $CH_4$ $1884.7 \pm 1$ ppb during September 2020 (https://gml.noaa.gov/obop/mlo/, last access: 15 April 2022). Data collected during this period from TCCON sites located in southern California (CIT and NASA Armstrong) were explored as background

sites; however, during this period, $X_{CO}$ was elevated due to local wildfires in those areas and thus not appropriate to use during this time.

## 3   Results

### 3.1   Observations of $X_{CO}$, $X_{CO_2}$, $X_{CH_4}$, and AOD from wildfires in the San Joaquin Valley

The first week of trace gas measurements is shown in Fig. 2, in addition to the daytime fire radiative power (FRP), an indicator of fire intensity measured by the Visible Infrared Imaging Radiometer Suite (VIIRS) active fire and thermal anomalies product from NOAA-20. Fire-emitted CO can be observed in the time series, and $X_{CO}$ is exceptionally high on 12 September, reaching mixing ratios 10 times higher than the previous days. A regional smoke plume was captured by the NOAA-20 VIIRS satellite on 12 September that originated from the SQF Complex and traveled west directly over the measurement site, as seen in Fig. 1b. The measurement on 12 September also corresponds to the highest FRP during this record. On the next day, 13 September, both fires remained active; however, their smoke plumes were transported northward, as reflected by a lower $X_{CO}$ in our observations relative to 12 September.

$X_{CO_2}$ and $X_{CH_4}$ were also enhanced on the 12 September smoke event and followed the same trend as $X_{CO}$ over the course of the day. Over 30 dairy farms are located northwest of the measurement site, and they are expected to influence observed $X_{CH_4}$ and $X_{CO_2}$; moreover, dairy influence is notable on days with predominantly westerly winds (e.g., 8 and 11 September). $X_{CO}$, $X_{CO_2}$, and $X_{CH_4}$ averaged at $154 \pm 78$ ppb, $413 \pm 1$ ppm, and $1938 \pm 27$ ppb from 8 September to 17 October. $X_{CO}$ and $X_{CO_2}$ peaked on 12 September at 1012.8 ppb and 421.6 ppm, while $X_{CH_4}$ peaked on 28 September at 2050.1 ppb due to the dairy farms in the area. The measured $X_{CO}$ on 12 September is the highest reported $X_{CO}$ value in the EM27/SUN literature. Retrievals of $X_{gas}$, using the EM27/SUN in such dense smoke plumes, has not been reported in previous studies. Using this date as a case study, we calculate total column EFs and MCE (further described in Sect. 3.4). We isolate the 12 September fire smoke plume by taking the $X_{CO}$ mixing ratios that exceeded the 98th percentile ($> 335.1$ ppb) from all observations over our measurement period. This period corresponded to mixing ratios recorded after 12:00 PDT, when $X_{CO}$ and $X_{CO_2}$ began to increase considerably.

The time since the emission of the observed smoke plumes was estimated to be $\sim 1.5$ h. This was calculated by dividing the distance away from the SQF Complex fire ($\sim 60$ km) by the average wind speed ($11.2 \pm 0.8$ m s$^{-1}$) at the height of the smoke plume ($4.1 \pm 1.2$ km). The height of the plume was determined by taking a mean of the available pixels within the smoke plume of aerosol layer height product from TROPOMI (http://www.tropomi.eu/data-products/

aerosol-layer-height, last access: 15 July 2022). The mean wind speed measured at $4.1 \pm 1.2$ km came from a 915 MHz wind profiler located in Visalia, CA, about 20 km west of the observational site (data available at ftp://ftp1.psl.noaa.gov/psd2/data/realtime/Radar915/, last access: 15 July 2022).

We show a time series of AOD at 500 nm derived for the first week of measurements in Fig. 2e (8–15 September), plotted with AOD at 500 nm from an AERONET station in Fresno, located about 90 km north of the measurement site (Fig. 1). Similar to observations of $X_{CO}$, enhancements of AOD are observed throughout the week, with the highest recorded AOD on 12 September. The observational sites were relatively far from each other ($\sim 90$ km), and although smoke reaching the two sites varied over these spatial scales, the FTIR AOD follows the same inter-day trend as the AOD measured by the AERONET, with a peak in AOD on 12 September. Intraday variability between the sites does not seem to follow the same trend. This suggests that the EM27/SUN AOD estimate was also able to qualitatively capture the increase in aerosols in the SJV as fires burned more intensely and smoke from fires moved into the valley due to synoptic conditions. A comparison between the FTIR and AERONET hourly AOD can be found in Fig. S3, where we find a slope of $1.4 \pm 0.3$ and $R^2$ of 0.39. Differences are observed in the AOD time series, as these two sites were downwind of two different fires in the Sierra Nevada. The Creek Fire was located directly east of Fresno, and the SQF Complex was located directly east of the EM27/SUN measurement site. This may be the reason why the peaks observed at the FTIR site are not seen in the Fresno AERONET data. Ahangar et al. (2022) determined that the SJV air quality was mainly impacted during the 8–15 September period, with the Creek and SQF Complex fires responsible for the majority of the smoke within SJV. Although the Creek Fire began on 5 September, the air quality began to deteriorate a few days after, possibly due to the westerly downslope winds that pushed the smoke east of the Sierra Nevada at the beginning of the fire (Cho et al., 2022). Low AOD from AERONET was observed prior to 8 September, with values of $0.50 \pm 0.28$, illustrating the air quality was cleaner and deteriorated after the activity from the Creek and SQF Complex fires increased (Ahangar et al., 2022).

Figure 3 shows $X_{CO}$ plotted against simultaneously collected AOD at 500 nm for 1 min intervals. The points are colored to distinguish the different measurement days from 8 to 15 September. The error bars are the uncertainty in AOD (further described in Appendix B), and for $X_{CO}$, it is 1 standard deviation (SD) on the mean. Due to the rapidly changing $X_{CO}$ as the fire plume traversed over the instrument, we use the standard deviation from the 1 min $X_{CO}$ mean as the natural variability or uncertainty, which is larger than the $X_{CO}$ instrument error. Table 1 shows the slope and intercepts, with standard errors from a York linear regression fit that considers errors in $x$ and $y$. We find strong relationships ($R^2 > 0.61$) between the EM27/SUN $X_{CO}$ and AOD at

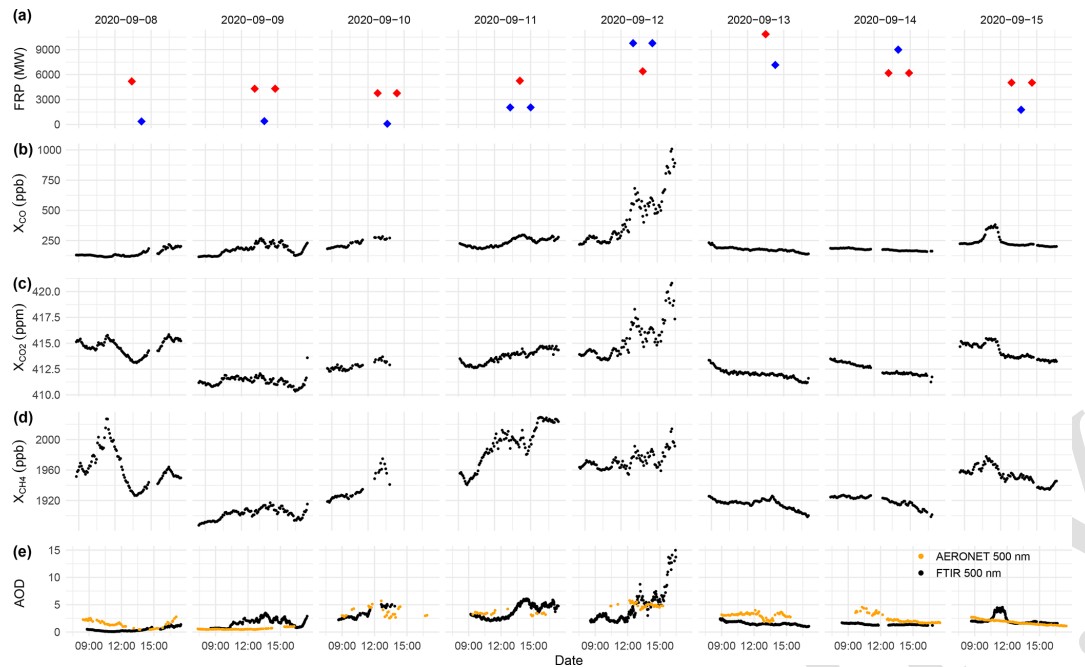

**Figure 2.** Time series during the first week of measurements for 8–15 September 2020. **(a)** Daytime total FRP from VIIRS NOAA-20 of Creek Fire (red) and SQF Complex (blue). **(b–d)** The 5 min mean observations from the ground-based EM27/SUN solar-viewing spectrometer of $X_{CO}$, $X_{CO_2}$, and $X_{CH_4}$. **(e)** FTIR-derived AOD (black) and AERONET AOD at 500 nm (orange).

500 nm, with slopes ranging from 29.01 to 92.41 ppb/AOD (Table 1 and Fig. 3). Several studies have also found a strong correlation between CO and AOD at 500 and 550 nm from fire events and downwind of polluted sources (Lobert, 2002; Paton-Walsh et al., 2005; Kampe and Sokolik, 2007; McMillan et al., 2008). McMillan et al. (2008) found mean slopes of fire plume observations from the Atmospheric Infrared Radiation Sounder (AIRS) CO and Moderate Resolution Imaging Spectroradiometer (MODIS) AOD that ranged from 40 to 74 ppb AOD and over clean regions slopes averaged at 35 ppb AOD. Most of the days in our observations have slopes that fall within these ranges, and the days with lower slopes (10 and 11 September) follow a similar linear trend (gray line in Fig. 3), as in McMillan et al. (2008), over a clean region in Alaska and Canada. Kampe et al. (2007) found that AOD or CO slopes varied strongly, and this variation may depend on age of smoke plume, distance from source, combustion efficiency, and local meteorological factors. Our measurements were sensitive to nearby smoke plumes in addition to mixed smoke from distant fires. The intercepts of the fitted lines reflect different local backgrounds of CO during measurement periods, with 10–12 September having the largest backgrounds of $X_{CO}$. We find that the AOD on 12 September reached values above 15, indicating extremely high aerosol loading from the smoke plume event transported from the SQF Complex in the Sierra Nevada.

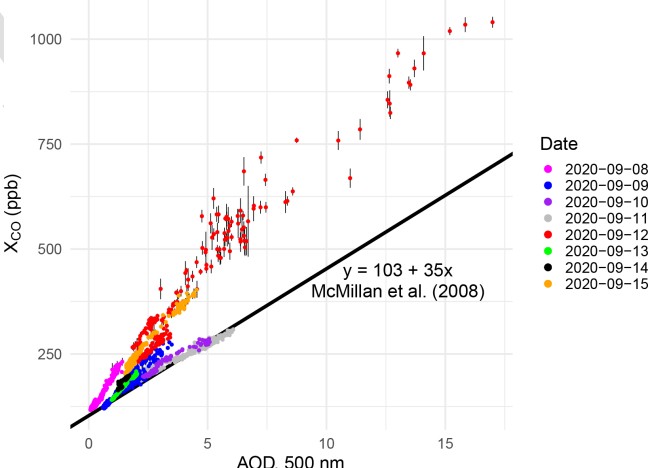

**Figure 3.** Scatterplot correlations of $X_{CO}$ and AOD at 500 nm from the FTIR for each day from 8–15 September 2020. Some days fall along the gray line that was derived from previous remotely sensed $X_{CO}$ to AOD relationships over a clean region. The red markers correspond to 12 September, the day of highest fire influence in our record.

## 3.2 Comparison of EM27/SUN and TROPOMI retrievals

In this section, we compare $X_{CO}$ retrieved from ground-based EM27/SUN observations downwind of the Sierra Nevada wildfires to satellite-based $X_{CO}$ retrievals from coincident TROPOMI overpasses. Previous studies of $X_{CO}$

**Table 1.** Summary of the York linear fit of $X_{CO}$ and AOD for 8–15 September 2020.

| Measurement date | Slope (ppb/AOD) | Intercept (ppb) | $R^2$ |
|---|---|---|---|
| 8 September 2020 | $81.8 \pm 0.2$ | $109.89 \pm 0.08$ | 0.97 |
| 9 September 2020 | $55.57 \pm 0.07$ | $87.47 \pm 0.09$ | 0.94 |
| 10 September 2020 | $33.84 \pm 0.04$ | $116.3 \pm 0.1$ | 0.94 |
| 11 September 2020 | $29.01 \pm 0.02$ | $128.60 \pm 0.07$ | 0.98 |
| 12 September 2020 | $62.42 \pm 0.07$ | $114.5 \pm 0.2$ | 0.94 |
| 13 September 2020 | $57.55 \pm 0.04$ | $90.94 \pm 0.06$ | 0.87 |
| 14 September 2020 | $92.41 \pm 0.06$ | $50.72 \pm 0.08$ | 0.61 |
| 15 September 2020 | $72.87 \pm 0.04$ | $95.60 \pm 0.08$ | 0.98 |

and $X_{CH_4}$ comparisons between TROPOMI and EM27/SUN have used TROPOMI soundings between 50 and 100 km from the observational site and used EM27/SUN measurements between 40 min and 1 h of the TROPOMI overpass as coincident criteria (Jacobs, 2021; Sha et al., 2021; Alberti et al., 2022b; Sagar et al., 2022). Given the spatial and temporal heterogeneity in smoke plumes from wildfires observed in Figs. 1 and 2, we perform a sensitivity study of different radii (10, 15, 20, 30, 40, and 50 km) from our observational site and time averages (10, 15, 20, and 30 min) to determine adequate criteria for comparison during a wildfire event. An illustration of the sensitivity analysis is shown in Fig. D1 in Appendix D.

We quantify the sensitivity of different TROPOMI radii and averaging times in comparison with our EM27/SUN observations by calculating the mean difference, mean relative difference, and $R^2$ between the linear regression fits for the measurements. We find that all combinations produce a positive mean bias, meaning that TROPOMI overestimates $X_{CO}$ compared to the EM27/SUN measurements. TROPOMI pixels within a radius of 5 km averaged with 30 min aggregations of EM27/SUN give the lowest mean difference of 10.64 ppb, a mean relative difference of 5.5 %, and the highest correlation coefficient of 0.99; however, only four points coincide during the measurement period. To maximize the number of coincidences while maintaining a low bias, we select 15 km as the maximum radius with a 30 min averaging time. This gives a total of 19 coincident data points and a mean difference of 17.2 ppb, a mean relative difference of $9.7 \pm 1.3$ %, and $R^2$ of 0.97. A time series of the coinciding data pairs from the EM27/SUN 30 min average observation period with TROPOMI overpass with 15 km radii are shown in Fig. 4a, and the comparison is shown in Fig. 4b. Applying these spatial and temporal criteria results in large variance for the largest measured $X_{CO}$ due to heterogeneity in the smoke plume event. The EM27/SUN displays a larger variance than TROPOMI due to capturing the 30 min temporal variability in the plume as it was transported above the instrument. We find a York linear regression fit of $y = (1.35 \pm 0.01)x - 39.30 \pm 0.58$. The mean relative difference

found in this study of $9.7 \pm 1.3$ % is similar to the systemic difference of $9.22 \pm 3.45$ % between TROPOMI and all TCCON stations (Sha et al., 2021). These results suggest that the differences found between the TROPOMI and EM27/SUN observations during wildfires are consistent with the systematic differences that exist between the two instruments; however, based on our sensitivity study, biases may exist based on sampling conditions in a spatially and temporally heterogenous source.

### 3.3 Emission factors and modified combustion efficiency

Emission ratios of CO and $CH_4$ on 12 September were calculated with respect to $CO_2$ (Fig. 5). $ER_{CO}$ was $0.1161 \pm 0.0005$, and the $ER_{CH_4}$ was $0.00730 \pm 0.00007$, resulting in an $EF_{CO_2}$ of $1632.9 \pm 163.3$ g $CO_2$ kg$^{-1}$ biomass combusted, $EF_{CO}$ of $120.5 \pm 12.2$ g CO kg$^{-1}$ biomass combusted, and $EF_{CH_4}$ of $4.3 \pm 0.8$ g $CH_4$ kg$^{-1}$ biomass combusted. We compared findings from our measurements to the literature values in temperate coniferous forest studies from the Sierra Nevada (Fig. 6) and other locations in North America (summarized in Table 2). All the studies listed in Table 2, except for this study, were based on aircraft measurements for temperate coniferous forests. Due to combustion-phase variability in field studies, we compare our atmospheric-column-based EFs in Fig. 6 with the most relevant studies from the Sierra Nevada, which shows our calculated values are within the expected linear range from in situ aircraft studies. The measurement uncertainties for the EFs were calculated by propagating the error from the ER linear regression standard error, $C_T$, and 10 % error from $M_{Biomass}$.

The average MCE for the smoke plume on 12 September was $0.89 \pm 0.21$ ($1\sigma$), meaning that observations of the smoke plume consisted of a mixture of flaming and smoldering combustion phases (Fig. 6). During the flaming phase of a fire, $CO_2$ is produced, and convection is created by high flame temperatures, producing the lofting of smoke. High-altitude smoke can be transported large distances, corroborated by observations of ash falling from the sky at a measurement site $\sim 60$ km away from the fire and clearly observable by satellite imagery (Fig. 1b). In contrast to the flaming phase, smoldering fires burn at lower intensity, and incomplete combustion side products like CO, $CH_4$, and organic carbon aerosol are produced. The MCE calculated from total column observations is averaged over the entire vertical plume, as it was being transported over the measurement site. The advantage of a plume-integrated MCE is that vegetation is burned differently throughout the fire, and the atmospheric column observations can represent the fire as a whole by integrating the smoke plume heterogeneity in the vertical atmospheric column.

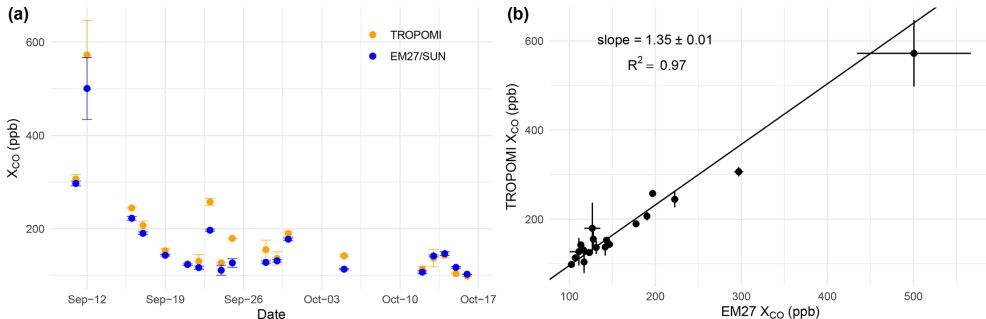

**Figure 4. (a)** Time series of coinciding EM27/SUN 30 min average observation period with a TROPOMI overpass with a 15 km radius. **(b)** Correlation between coinciding TROPOMI and EM27/SUN data pairs. The error bars are the standard deviation of the TROPOMI-averaged pixels at 15 km and the EM27/SUN 30 min observation.

**Table 2.** Summary from past airborne studies and the present study of modified combustion efficiency (MCE) and emission factors (EFs; $g\,kg^{-1}$) of CO and $CH_4$ for temperate coniferous forests in North America and the Sierra Nevada.

| Studies | | MCE | $EF_{CO_2}$ | $EF_{CO}$ | $EF_{CH_4}$ |
|---|---|---|---|---|---|
| North America | | | | | |
| Radke et al. (1991)* | Northwestern USA coniferous forest | 0.919 | 1641 | 93 | 3.03 |
| Yokelson et al. (1999)* | Southeastern USA pine forest understory | 0.926 | 1677 | 86 | – |
| Yokelson et al. (2011) | Mexico rural pine–oak forests | 0.908 | 1603 | 103 | 3.66 |
| Burling et al. (2011)* | Conifer forest understory in southeastern USA and Sierra Nevada mountains | $0.936 \pm 0.024$ | $1668 \pm 72$ | $72 \pm 26$ | $3.0 \pm 2.4$ |
| Urbanski et al. (2013) | Rocky Mountains conifer forest fires | 0.85–0.92 | 1527–1681 | 89.3–173 | 4.4–12.1 |
| Liu et al. (2017) | Western USA mixed conifer wildfires | 0.912 | $1454 \pm 78$ | $89.3 \pm 28.5$ | $4.9 \pm 1.5$ |
| Sierra Nevada | | | | | |
| Burling et al. (2011) | Turtle fire* (11 November 2009) | 0.91 | 1599 | 97 | 5.51 |
| | Shaver fire* (10 November 2009) | 0.885 | 1523 | 126 | 7.94 |
| Yates et al. (2016) | Rim fire (26 August 2013) | 0.94 | $1675 \pm 285$ | $92.5 \pm 16$ | $4.8 \pm 0.8$ |
| | Rim fire (29 August 2013) | 0.94 | $1711 \pm 292$ | $69.5 \pm 12$ | $4.7 \pm 0.8$ |
| | Rim fire (10 September 2013) | 0.88 | $595 \pm 272$ | $138.4 \pm 24$ | $7.5 \pm 1.3$ |
| Liu et al. (2017) | Rim fire (26 August 2013) | 0.923 | $478 \pm 11$ | $78.7 \pm 4$ | $4.43 \pm 0.25$ |
| This study | SQF Complex fire (12 September 2020) | $0.89 \pm 0.21$ | $1632.9 \pm 163.3$ | $120.5 \pm 12.2$ | $4.3 \pm 0.8$ |

* Includes prescribed burns.

## 3.4 Enhancement ratios of livestock and wildfire emissions

The EM27/SUN's location enabled us to sample transient fire plumes from local and state wildfires but was also located near a large cluster of dairy farms, which are a large regional source of $CH_4$ emissions (Heerah et al., 2021; Marklein et al., 2021). Dairy farms are known to emit significant amounts of $CH_4$ from the animals' enteric fermentation and on-farm manure management. Because fires also emit $CH_4$, we explored whether dairy and fire sources in this region can be disentangled using the enhancement ratios of the different species measured by the EM27/SUN. Enhancement ratios are also known as the normalized excess mixing ratios. Excess mixing ratios are calculated by subtracting the mixing ratio of a species from a source plume minus a mixing ratio of the same species in background air. To correct for dilution, excess mixing ratios are normalized by a stable tracer

such as $CO_2$. When an enhancement ratio does not change with dilution and mixing with background air, then the enhancement ratio is equal to the emission ratio (ER) of a source (Yokelson et al., 2013). Furthermore, our measured $\Delta X_{CH_4}/\Delta X_{CO_2}$ ratios enable us to investigate the contribution of state wildfires to $CH_4$ emissions in 2020. To constrain the observed enhancements, we compared the enhancement ratios of $\Delta X_{CH_4}/\Delta X_{CO_2}$ from September–October 2020 to enhancement ratios collected in September 2018 and 2019 in the same local area that characterize non-fire years. September 2018 and 2019 measurements are further described in the Supplement. We focused on observation days with statistically significant correlations ($n = 26$ d) between $CH_4$ and $CO_2$ enhancements ($R^2 > 0.5$ and $p < 0.05$) to characterize the enhancement ratios of the SJV non-fire years.

During September–October 2020 observations, $\Delta X_{CH_4}/\Delta X_{CO_2}$ ratios of dairy farm influence were found on several days, in addition to lower slopes indicative of

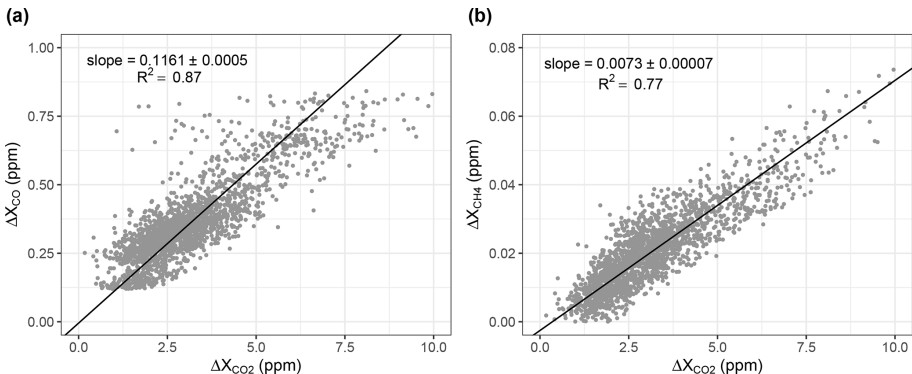

**Figure 5.** Relationship between **(a)** $\Delta CO$ and **(b)** $\Delta CH_4$ against $\Delta CO_2$ during the SQF Complex wildfire plume on 12 September 2020.

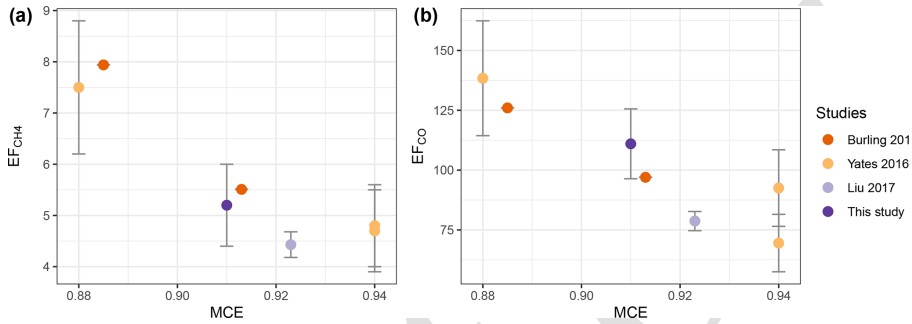

**Figure 6.** Emission factors $(\mathrm{g\,kg^{-1}})$ for **(a)** $CH_4$ and **(b)** CO as a function of MCE for temperate coniferous forests from Sierra Nevada wildfires.

combustion sources (Fig. 7; gray markers). The 12 September smoke plume event is highlighted in Fig. 7 (red markers) and has a smaller emission ratio of $7.3 \pm 0.07\,(\mathrm{ppb/ppm})$ compared to larger $\Delta X_{CH_4}/\Delta X_{CO_2}$ enhancement ratios of $38.4 \pm 21.7$ and $30.5 \pm 5.0\,(\mathrm{ppb/ppm})$ observed in September 2018 and 2019. Similar non-wildfire ratios of $\Delta X_{CH_4}/\Delta X_{CO_2}$ were found in Hanford, $\sim 40\,\mathrm{km}$ west of our observation site, from an aircraft study ranging from 35.9–44.4 (ppb/ppm) during a winter campaign (Herrera et al., 2021). Other column-based studies have determined the $X_{CH_4}/X_{CO_2}$ for urban sources in the Los Angeles city, finding ratios ranging from 6.65 to 9.96 (ppb/ppm) in 2015 (Chen et al., 2016) and $11 \pm 2\,‰$ in 2008 (Wunch et al., 2009). Wunch et al. (2009) determined that urban fossil fuel and wildfire $X_{CH_4}/X_{CO_2}$ ratios are very similar due to incomplete combustion, and the ratios are not distinct enough to separate. In the vicinity of the measurement site in the SJV, there is a strong influence of dairy farm agriculture and minimal urban emissions away from population centers; thus, we are able to separate $\Delta X_{CH_4}/\Delta X_{CO_2}$ from dairy sources, from fire, or from possible urban emissions. The $CH_4/CO_2$ enhancement ratios observed in this area make it evident that dairy farm operations are the dominant source of $CH_4$ during fire and non-fire days. Nevertheless, $CH_4$ enhancements during the strong smoke events greatly

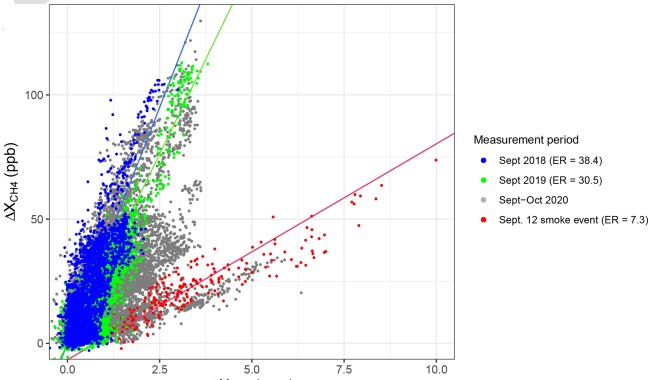

**Figure 7.** Correlation plots of $\Delta X_{CH_4}$ vs. $\Delta X_{CO_2}$ for SJV measurements collected during non-fire years in September 2018 (blue) and 2019 (green) and during the fire period of September–October 2020 (gray). The 12 September 2020 smoke event (red), highlighted with a linear fit through the day's data, clearly shows a distinct $\Delta X_{CH_4}/\Delta X_{CO_2}$ relationship.

exceeded $CH_4$ enhancements from local dairy sources on hourly timescales.

## 3.5 Total methane emissions from wildfires in California

The immense scale of the 2020 wildfires meant that they released a significant amount of $CO_2$ emissions, equivalent to about 36 % of the state's $CO_2$ budget for the year (CARB, 2022a). Our observations of $X_{CH_4}$ suggest that the wildfires may also have had a significant effect on the state's $CH_4$ budget. Given the importance of reducing $CH_4$ emissions for meeting California's climate goals, we calculate the amount of $CH_4$ released from wildfires that burned in the state in 2020 by using estimates of $CO_2$ emissions from the state's wildfire inventory, the $ER_{CH_4}$ calculated from our study, and from the literature values.

The California Air Resources Board (CARB) reported that a total of 106.7 Tg of $CO_2$ was emitted from 2020 wildfires and reported individual $CO_2$ emission estimates for the 20 largest wildfires of 2020. Using the reported $CO_2$ emission estimates from individual wildfires, we derived $CH_4$ emissions by multiplying the $CO_2$ estimates with emission ratios of $CH_4$ ($ER_{CH_4}$) from wildfire smoke and molecular mass ratios as follows:

$$E_{CH_4} = \left(ER_{CH_4} \times \frac{M_{CH_4}}{M_{CO_2}}\right) E_{CO_2}, \tag{5}$$

where $E_{CH_4}$ is the emissions of $CH_4$ in $Gg\,yr^{-1}$, $ER_{CH_4}$ is the emission ratio of $CH_4$ with respect to $CO_2$ (in $mol\,mol^{-1}$), $M_{CH_4}$ is the molar mass of $CH_4$, $M_{CO_2}$ is the molar mass of $CO_2$, and $E_{CO_2}$ are the individual wildfire emissions of $CO_2$ in $Gg\,yr^{-1}$. Emission ratios from fires are dependent on vegetation type; fires in California fell into temperate forest, shrubland or grassland vegetation types (Xu et al., 2022). Based on the generic vegetation classification from the Fire INventory from NCAR (FINN) model (https://www.acom.ucar.edu/Data/fire/, last access: 15 July 2022), we classify the top 20 California wildfires of 2020 into the three types, based on the dominant vegetation of temperate forest, shrublands, or grasslands. We assign an $ER_{CH_4}$ for each general vegetation type based on the mean $EF_{CH_4}$ found in Xu et al. (2022) that summarized EFs from Prichard et al. (2020). The standard deviation of the $EF_{CH_4}$ was calculated based on Prichard et al. (2020) and taken as the uncertainty that was then propagated in the $ER_{CH_4}$ calculations. For the Sierra Nevada wildfires (Creek Fire, SQF Complex, and North Complex), we derive $ER_{CH_4\_avg}$ by first calculating an average $EF_{CH_4\_avg}$ from the Sierra-Nevada-specific $EF_{CH_4}$ in Table 1 ($EF_{CH_4\_avg} = 5.6 \pm 1.5\,g\,kg^{-1}$). We then used Eq. (2) to solve for $ER_{CH_4}$ with $C_T$ equal to 1 ($ER_{CH_4\_avg} = 0.0084 \pm 0.0022$). A summary of $ER_{CH_4}$ can be found in Table E1 (Appendix E). Methane emissions for the top 20 wildfires were then calculated, using Eq. (5) with CARB's $CO_2$ estimate from each individual fire, and summed to obtain a total $CH_4$ emitted from these reported wildfires. Figure 8 shows the estimated $CH_4$ emissions from the top 20 wildfires of 2020 compared to CARB's 2020 anthropogenic $CH_4$ inventory emissions (CARB, 2022b). The

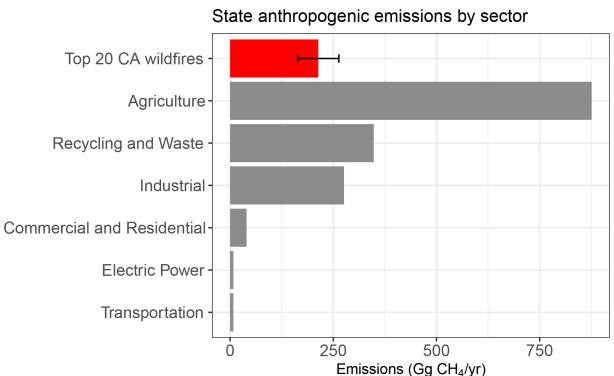

**Figure 8.** California $CH_4$ emissions from 2020 calculated for the top 20 wildfires of 2020 compared to the state's anthropogenic $CH_4$ emissions from the 2020 inventory (CARB, 2022b). The industrial sector also includes oil and gas emissions.

error bar from this estimate was calculated by propagating the general vegetation $ER_{CH_4}$ error from Table E1 into each individual wildfire $CH_4$ estimate and adding errors in quadrature to obtain a total error. CE2 The 20 largest wildfires of 2020 emitted 92 % of the total $CO_2$ emissions released from wildfires in that year and emitted $213.7 \pm 49.8\,Gg\,CH_4$ or 13.7 % of total anthropogenic $CH_4$ emissions.

## 4 Discussion

We demonstrate that EM27/SUN total column measurements can be used for calculating MCE and EFs in smoke plumes transported from wildfires, especially for high-altitude smoke, adding important new estimates for fires in this region. For the Sierra Nevada, only three field-based studies have estimated emission factors in this area, despite the increase in wildfire burns over the previous decade (Burling et al., 2011; Yates et al., 2016; Liu et al., 2017). Table 1 highlights the variety of EFs and MCE sampled over the Sierra Nevada and North America. Despite the variability, our emission factor estimates from the 12 September event for $CO_2$ ($1632.9 \pm 163.3\,g\,kg^{-1}$), CO ($120.5 \pm 12.2\,g\,kg^{-1}$), and $CH_4$ ($4.3 \pm 0.8\,g\,kg^{-1}$) are within the range of those reported from the Sierra Nevada conifer forests. Additionally, our calculated emission factors also agree well with recently compiled emission factors for North American conifer forests, where Prichard et al. (2020) found a fire average for $EF_{CO_2}$ of $1629.54 \pm 63.43$, $EF_{CO}$ of $104.01 \pm 34.93$, and $EF_{CH_4}$ of $5.05 \pm 2.41$. Methane emission ratios reported for smoldering fires that were characterized by direct $O_2/CO$ measurements for the California 1999 Big Bar Complex fire are also consistent with our measurements (Lueker et al., 2001). These atmospheric-column-based estimates contribute to the limited number of EFs for temperate forests and are particularly important given the scale of the fires that occurred in 2020 in California. Empirically quantified EFs in

temperate conifer forests are limited in number, and many of the measurements in these regions are from prescribed burning for land management (Burling et al., 2011; Akagi et al., 2011; Urbanski, 2013). Because prescribed burns typically occur during favorable atmospheric conditions, with specified fuel, and during non-wildfire seasons, it is possible that prescribed burn EFs may not represent wildfire EFs that burn under different conditions favorable to wildfires (Urbanski, 2013). There is a need for biome-specific EFs to quantify the amount of trace gas or aerosol emitted per kilogram of biomass burned, and these EFs are essential model inputs for estimating total greenhouse gas and aerosol emissions of fires.

While the advantages of this technique allow for understanding regional-scale emissions, limitations exist with this method. The EM27/SUN solar column observations are limited to the daytime hours, as the instrument requires the Sun as the light source. For this reason, we were not able to capture nighttime observations despite the continued release of smoke emissions and the growing concern of increasing nighttime wildfire activity in the continental USA (Freeborn et al., 2022). Additionally, optically thick smoke plumes obstruct the sunlight and prohibit continued measurements when the solar disk is not traceable by the instrument's solar tracker. Exposing the instrument's mirrors to harsh conditions such as ash decreases the instrument signal and may decrease the lifetime of the mirrors. Although total column measurements are sensitive to larger scales than in situ stations, the FTIR is limited to the line of sight of the instrument and on occasion can miss the plume, like we did on 13 and 14 September, whereas aircraft observations have extensive spatial reach and more flexibility in locating and sampling plumes to obtain spatially rich information of the plume. However, when used in tandem with satellite observations, our instrument collects continued temporal observations of a site of interest that a satellite does not; thus, synchronous observations provide a better spatiotemporal understanding of the emission source. EM27/SUN instruments are also costly, which can limit the number of instruments deployed. Unless instruments are secured properly, as has been done in long-term FTIR network studies (Frey et al., 2019; Dietrich et al., 2021), measurements require personnel to set up and operate the instrument daily. The EFs, MCE, and their uncertainties fall within the range of expected values, thus lending confidence that this technique can be used for studying combustion phases of wildfires for other vegetation types. Despite the limitations of the EM27/SUN, we demonstrate the ability to gather new information of EFs, MCE, and AOD for understudied vegetation types and regions. Furthermore, the EM27/SUN observations can be used as a validation tool for orbiting satellites like TROPOMI, the Orbiting Carbon Observatory-2 (OCO-2), OCO-3, and future satellites. The next-generation weather forecasting, greenhouse gas, and air pollutant satellites such as TEMPO (Tropospheric Emissions: Monitoring of Pollu-

tion) will have more temporal frequency and greater spatial resolution, allowing for continuous monitoring of burning activity and smoke emissions (Zoogman et al., 2017). This may allow remote sensing products to provide new insight into the fuel properties of many types of vegetation in remote areas. It will also be important to evaluate satellite-based observations with ground-based stations like the EM27/SUN, as we did in this study.

Simultaneous measurements of ground-based total columns and satellites allow for a spatial and temporal understanding of the fire events. The $X_{CO}$ enhancement from the 2020 wildfires in the Sierra Nevada was also observed from space, and in concentrated smoke plumes, $X_{CO}$ was up to 10 times higher than the local background, which was clearly visible in the TROPOMI soundings during the 12 September smoke plume event. Pairing stationary ground-based column observations with satellites can help in understanding regional wildfires at a greater spatial and temporal scale. Although TROPOMI has daily global coverage with high spatial resolution, daily snapshots are often not enough to understand the behavior of a fire. Conversely, stationary, ground-based instruments are limited to observing a line of sight or point in space. As an instrument with the capability of measuring atmospheric columns, the EM27/SUN can help close the gap in the temporal scale of satellite observations. The EM27/SUN measured continuously in the daytime, filling in the temporal gaps from the satellite TROPOMI's single overpass observations. A sensitivity study showed that a smaller radius of 5 or 15 km from TROPOMI observations, paired with 30 min averaging around the overpass time, gave better statistical agreement during wildfire events. This strong correlation of $X_{CO}$ between TROPOMI and the EM27/SUN has been observed before in urban sites (Sagar et al., 2022; Alberti et al., 2022b) and in rural Alaska (Jacobs, 2021). Jacobs (2021) found that wildfire influences in $X_{CO}$ resulted in a high observational variance in EM27/SUN observations and suggested that this may be due to spatial and temporal variability in the smoke plume measured by TROPOMI and the EM27/SUN. The $9.7 \pm 1.3\%$ mean relative difference between the EM27/SUN and TROPOMI found in this study may also be due to the averaging of the smoke plume's heterogeneity within each TROPOMI comparison point. Alternatively, Rowe et al. (2022) found that multiple scattering on aerosols may be responsible for 5%–10% of the increased $X_{CO}$ observations from TROPOMI in thick smoke plumes.

The air quality index in the SJV was at an all-time high in the hazardous range for weeks during the 2020 wildfire season (Morris and Dennis, 2020), and AOD at the AERONET site in Fresno, the yearly average from 2002–2019 increased by 3 to 5 times (Cho et al., 2022). FTIR-derived AOD at 500 nm reached extremely high levels during the 12 September smoke plume event and followed the same trend on other days as the trace gas enhancements. The slopes during low

smoke and high smoke days were consistent with previous satellite observations by McMillan et al. (2008). Previously, simultaneous measurements of aerosols and trace gases from the same instrument have been limited due to the aerosol burden interfering with retrieval of trace gases. For example, the majority of the TROPOMI $X_{CH_4}$ product was flagged out completely near the observational site during the 7–15 September period and hence was not included in this analysis. The EM27/SUN demonstrated the potential to elucidate trace gas and aerosol relationships even during thick aerosol periods. Similarly, future studies may use simultaneous measurements from the TROPOMI $X_{CO}$ product and AOD to study the regional impacts from wildfires (Chen et al., 2021). Scattered diffuse light during high aerosol loading from biomass burning may decrease the reliability of the AOD observations; thus, further verification of the FTIR-derived AOD during high aerosol loading is required. Since the nearest AERONET station was relatively far away from our EM27/SUN site, we cannot do a true side-by-side comparison. However, the FTIR-derived AOD showed the same baseline pattern as the AERONET site in Fresno, demonstrating the ability of the EM27/SUN to simultaneously measure AOD and trace gases through a thick plume of smoke, which can elucidate mechanisms within smoke plumes.

Estimates of $CH_4$ emitted from biomass burning are commonly calculated for global inventories such as FINN, the Global Fire Emissions Database (GFED), and the Intergovernmental Panel on Climate Change (IPCC) guidelines that rely on satellite observations of the area burned and observation-based emission factors (Wiedinmyer et al., 2011; van der Werf et al., 2017); however, these bottom-up $CH_4$ inventories tend to report large uncertainties (Saunois et al., 2020). In California, statewide wildfire estimates of $CO_2$ and PM are based on the USA Forest Service's First Order Fire Effects Model (FOFEM; Reinhardt and Dickinson, 2010), though reports of $CH_4$ estimates from wildfires are lacking, despite the importance of $CH_4$ for meeting the state's ambitious climate goals. To reduce the uncertainties and constrain emissions of global and local $CH_4$ budgets, more atmospheric-based estimates of $CH_4$ emissions are required, as currently only a few observation-based studies exist that focus on estimating $CH_4$ emissions (e.g., Mühle et al., 2007; Worden et al., 2013). As wildfires become more frequent with climate change, monitoring trace gases and particulates may become especially challenging in mixed source areas like the San Joaquin Valley where concentrations can become amplified by stagnant atmospheric conditions. Moreover, the fire-added $CH_4$ may hamper the evaluation of greenhouse gas emission reduction initiatives at the state scale and at the global scale by adding unaccounted for $CH_4$ to the atmosphere. Using CARB's 2020 wildfire emission estimate for $CO_2$, we calculated the $CH_4$ contribution from the 20 largest fires to be $213.7 \pm 49.8\,\mathrm{Gg\,CH_4}$. These wildfires alone emitted 13.7 % of the total state anthropogenic $CH_4$ emissions, which is more than the commercial and residential, transportation, and electric power sectors. While estimated $CH_4$ emissions from wildfires are smaller in magnitude than inventoried emissions from agriculture, waste, and industrial sources, this source should be accounted for in the state's greenhouse gas inventories, given its magnitude and the large impacts on the atmospheric $CH_4$ during wildfire periods. Globally, about 10 % of anthropogenic global $CH_4$ is emitted by biomass burning (Saunois et al., 2020) and may be an important and unaccounted for positive feedback to climate change, given the effect of increasing temperatures on fire severity.

## 5 Conclusions

Over the past 50 years, approximately three-quarters of the area burned by wildfires in California has been in the Sierra Nevada and North Coast, highlighting the importance of studying emission factors from fires in these ecosystems. However, there are surprisingly few observations of emission factors from these fires despite their importance for California's greenhouse gas budget and air quality implications. The ground-based EM27/SUN is a useful instrument for understanding the emissions of trace gases and aerosols from wildfires at regional scales. The portable nature of the EM27/SUN for deployment downwind of fires and for calculating important variables like EFs and MCE. Having alternative techniques to observe emissions of wildfires can help add to the limited number of emission factors for understudied vegetation and improve the emission estimates of biomass burning. Several studies have demonstrated the utility in FTIR-derived EFs for studying whole fire emissions from open-path instruments and vertically integrated measurements. Our total column MCE and EF with respect to $CO_2$ are the first to be reported from ground-based FTIR measurements in California.

Wildfire smoke produced overcast skies throughout the western USA during this period, with smoke plumes being transported over long distances. The EM27/SUN measures a vertically integrated regional signal but is limited spatially compared to observations from satellites. Here we show that a combination of the two can elucidate the spatiotemporal variability in wildfire emissions. We find a strong agreement between the EM27/SUN and TROPOMI, with a mean relative difference of $9.7 \pm 1.3$ % between the platforms. This is consistent with systematic differences between TCCON and TROPOMI, in addition to previous studies of EM27/SUN $X_{CO}$ for wildfires in rural Alaska and Idaho. Additionally, our solar spectral measurements at 1020.9 nm were used to derive AOD at 500 nm to compare to a nearby AERONET site and to compare AOD to CO ratios with previous studies. We found that our AOD values followed the same intraday pattern as the AERONET observations. AOD at 500 nm reached extreme levels of up to 15 during the smoke plume event. Good agreements were found in the AOD to CO ratios

with those observed over the USA and Canada from MODIS AOD and AIRS CO.

Finally, we find that a significant amount of $CH_4$ was emitted from the largest 20 wildfires of 2020 in California. Given the importance of the $CH_4$ emissions reduction for the state, our study suggests that wildfires are an important source of $CH_4$ for California and may delay the meeting of the state's ambitious goals for reducing greenhouse gas emissions. Atmospheric $CH_4$ emissions released during wildfire periods should also be accounted for in statewide greenhouse gas inventories, as wildfire $CH_4$ enhancements are clearly measurable, and their yearly emissions are comparable to or larger than other $CH_4$ sectors. Overall, our analysis contributes to the development of techniques for analyzing remotely sensed greenhouse gases and aerosol measurements from wildfires.

## Appendix A: EM27/SUN correction factors

The EM27/SUN was co-located with the CIT TCCON for 2–3 d before (2 and 3 September 2020) and after (30 and 31 October and 1 November 2020) the field measurements. A summary of the correction factors is shown in Table A1. An averaging kernel correction has been applied to the EM27/SUN observations prior to comparison, following Hedelius et al. (2016). Due to a camera misalignment on 2 and 3 September, $X_{CO}$ correction factors for those dates are not reported.

**Table A1.** Summary of correction factors from co-located EM27/SUN measurements with TCCON at CIT.

| $X_{gas}$ | 2 and 3 September | 30 and 31 October; 1 November |
|---|---|---|
| $X_{CH_4}$ | 0.9986 (0.0002) | 0.9976 (0.0001) |
| $X_{CO_2}$ | 1.0042 (0.0001) | 1.0036 (0.0001) |
| $X_{CO}$ | – | 0.9737 (0.0028) |
| $X_{H_2O}$ | 1.0044 (0.0005) | 1.0101 (0.0005) |

## Appendix B: Aerosol optical depth calculation

To calculate AOD from the EM27/SUN solar measurements, we follow the methods described in Barreto et al. (2020), who found good agreement between AERONET and TCCON FTIR-derived AOD at the high-altitude Izaña Atmospheric Observatory in Spain. Their analysis was performed on degraded TCCON FTIR solar spectra ($0.5\,cm^{-1}$) to assess the capability of lower-resolution FTIR EM27/SUN instruments to detect broadband aerosol signal. In total, 10 interferogram scans were co-added to increase the signal-to-noise ratio of the aerosol retrieval for a total integration time of 1 min. The uncertainty in the AOD product in this study is determined by adding in quadrature the estimated uncertainty of $\sim 0.006$, determined in Barreto et al. (2020), for the 10 co-added interferogram scans for a total uncertainty of 0.02 for a 1 min analysis. We calculated AOD from four

recommended micro-windows, with high solar transmission centered at 1020.9, 1238.25, 1558.25, and 1636 nm and compared the results with a nearby AERONET site located in Fresno, CA.

We apply the methods, further described in Barreto et al. (2020), that are based on the Beer–Lambert–Bougher attenuation law, as follows:

$$V_\lambda = V_{o,\lambda} \cdot d^{-2} \cdot \exp(-m \cdot \tau_\lambda), \tag{B1}$$

where $V_\lambda$ is the measured solar irradiance at wavelength $\lambda$, $V_{o,\lambda}$ is the spectral irradiance outside the Earth's atmosphere at wavelength $\lambda$, $d$ is the ratio of mean to actual Sun–Earth distance, and $m$ is the optical air mass (Kasten and Young, 1989). The $V_o$ is derived from the Langley method, by utilizing the measured solar intensity ($V$) vs. the optical air mass ($m$) and extrapolating to an optical air mass of zero. The total optical depth ($\tau_\lambda$) is the sum of the optical depth of Rayleigh scattering ($\tau_{R,\lambda}$), gas absorption ($\tau_{g,\lambda}$), and aerosols ($\tau_{a,\lambda}$), as follows:

$$\tau_\lambda = \tau_{R,\lambda} + \tau_{g,\lambda} + \tau_{a,\lambda}. \tag{B2}$$

Barreto et al. (2020) carefully selected and evaluated several FTIR micro-windows to minimize the gas absorption; thus, $\tau_{g,\lambda}$ is considered negligible. Rayleigh scattering is calculated, following Bodhaine et al. (1999), using the pressure measured at the measurement site by the ZENO Weather Station. The AOD $\tau_{a,\lambda}$ can then be calculated by subtracting the Rayleigh scattering from the equation below:

$$\tau_{a,\lambda} = \frac{\ln\left(V_{o,\lambda} \cdot d^{-2}\cdot\right) - \ln(V_\lambda)}{m} - \tau_{R,\lambda}. \tag{B3}$$

A cloud filter is applied to the spectra based on the measured fractional variation in solar intensity (fvsi). We set this quality filter to a maximum of 0.5 % variability to ensure minimum cloud interference. The optical air mass range for the Langley plot calibrations were performed from $1.5 \leq m < 7$ to avoid large errors at smaller air masses and a turbidity influence at solar noon. A plot of $\ln(V_o)$ (found in Fig. B1) displays the calculated $\ln(V_o)$ over time from September to November 2020. Mirror degradation and exposure to dust or ash from fires can be observed in a declining $\ln(V_o)$, and a sudden jump in $\ln(V_o)$ is observed in late October and early November after the mirrors were cleaned, suggesting that debris had diminished the solar intensity measured by the FTIR instrument. Due to the varying $\ln(V_o)$, we calculate AOD only for the first week of data collection (8–15 September), using the $\ln(V_o)$ obtained during the earlier period of September (summarized in Table B1).

A time series of the FTIR-derived AOD for the four micro-windows is shown in Fig. B2, where a spectral dependance of the aerosol absorption can be observed in the plot with longer wavelengths recording smaller AOD. Although our FTIR-derived AOD is limited to the spectral range from the

**Table B1.** Mean values of $\ln(V_o)$ from 14, 19, and 24 September 2020 used for deriving AOD.

| Microwindow (nm) | Mean $\ln(V_o)$ | SD | $n$ |
|---|---|---|---|
| 1020.9 | 15.17 | 0.11 | 3 |
| 1238.25 | 16.01 | 0.09 | 3 |
| 1558.25 | 16.34 | 0.08 | 3 |
| 1636 | 16.35 | 0.08 | 3 |

FTIR detector (1020.9–1636 nm), we used the Ångström exponent to derive the FTIR AOD at 500 nm to enable a comparison with other studies (shown in Fig. 3). A plot of AOD at 1020.9 and 1636 nm with AERONET at 1020 and 1640 nm can be found in Fig. B3.

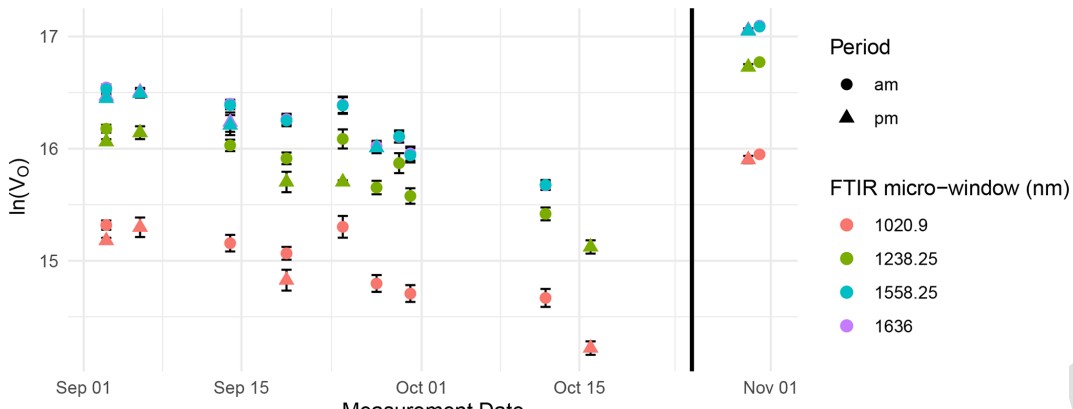

**Figure B1.** Absolute calibration for the Langley exponential analysis of the EM27/SUN solar spectra over time from September to November 2020. Mirrors became significantly dirtier and dustier over the course of the measurement period. The $\ln(V_O)$ increased considerably after the instrument mirrors were cleaned once the field campaign ended (black line).

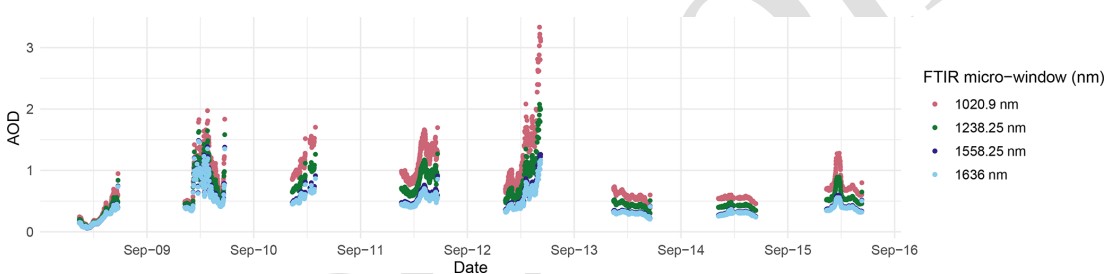

**Figure B2.** Time series of AOD for the four micro-windows from 8 to 15 September 2020.

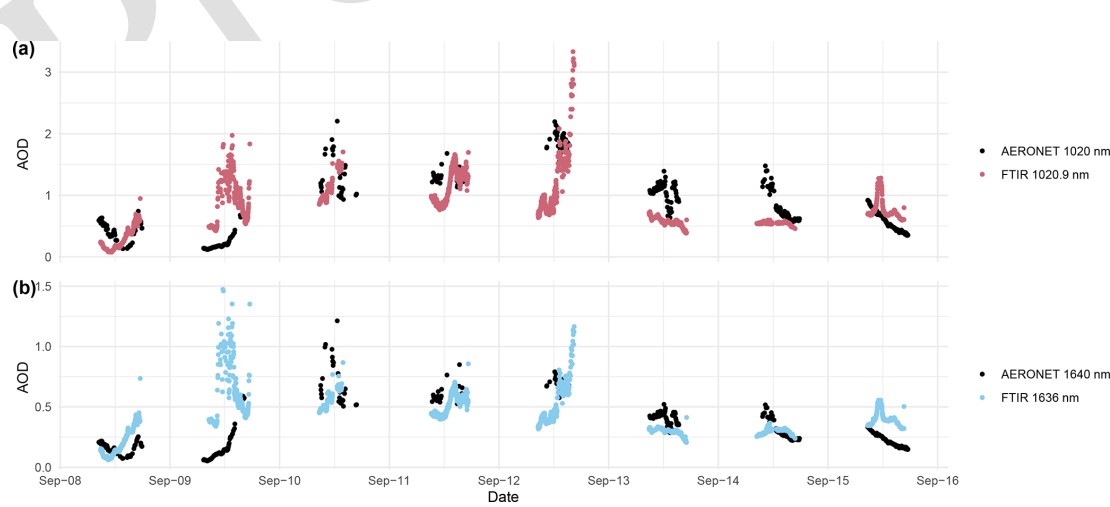

**Figure B3.** Time series of AOD from FTIR for the **(a)** 1020.9 nm (red) and **(b)** 1636 nm (blue) windows and AERONET (black) located in Fresno, CA, $\sim 90$ km north of the measurement site.

## Appendix C: EM27/SUN sensitivity

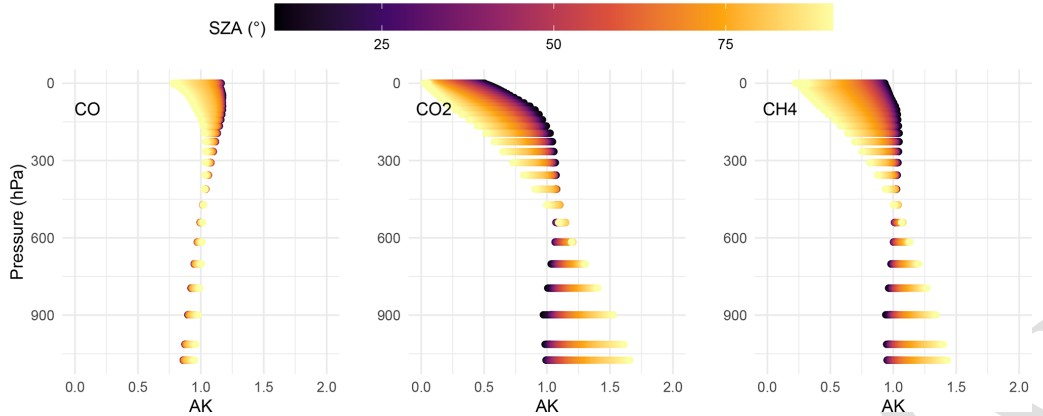

**Figure C1.** Averaging kernel (AK) of EM27/SUN of $X_{CO}$, $X_{CO_2}$, and $X_{CH_4}$ colored by solar zenith angle (SZA).

The EM27/SUN has different instrument sensitivities that are defined by the averaging kernels (AK) for each species measured (shown in Fig. C1). The difference in sensitivity for the trace gases may introduce a bias in the calculated ERs and MCE. Most of the difference is expected to be at the height of the plume where the smoke is concentrated at 4.1 km ($\sim 600$ hPa). Following the methods of Hedelius et al. (2018), we divide the enhancements of $\Delta X_{CO_2}$ and $\Delta X_{CO}$ by the averaging kernel at that smoke plume height as follows:

$$
\begin{aligned}
&\text{MCE}_{\text{AK corrected}}(\text{SZA}) \\
&= \frac{\Delta X_{CO_2}/\text{AK}(\text{SZA})_{CO_2,600\,\text{hPa}}}{\Delta X_{CO_2}/\text{AK}(\text{SZA})_{CO_2,600\,\text{hPa}} + \Delta X_{CO}/\text{AK}(\text{SZA})_{CO,600\,\text{hPa}}},
\end{aligned}
\tag{C1}
$$

where $\text{AK}_{600\,\text{hPa}}$ is the averaging kernel sensitivity for CO or $CO_2$ as a function of the solar zenith angle (SZA). The mean relative difference in the correction for the 12 September plume event is $-1.1\,\%$; thus, not applying this correction would overestimate the MCE by 1.1 %. Similarly, for the ERs, we correct the enhancements prior to fitting the points with a linear regression for the 12 September plume event:

$$
\text{ER}_{i,\text{AK corrected}} = \frac{\Delta X_i / \text{AK}(\text{SZA})_{i,600\,\text{hPa}}}{\Delta X_{CO_2} / \text{AK}(\text{SZA})_{CO_2,600\,\text{hPa}}}.
\tag{C2}
$$

Without applying this correction, $E_{CH_4}$ would be underestimated by 9.5 % and $E_{CO}$ by 14.2 % due to the difference in sensitivity.

## Appendix D: TROPOMI and EM27/SUN coincident criteria sensitivity analysis

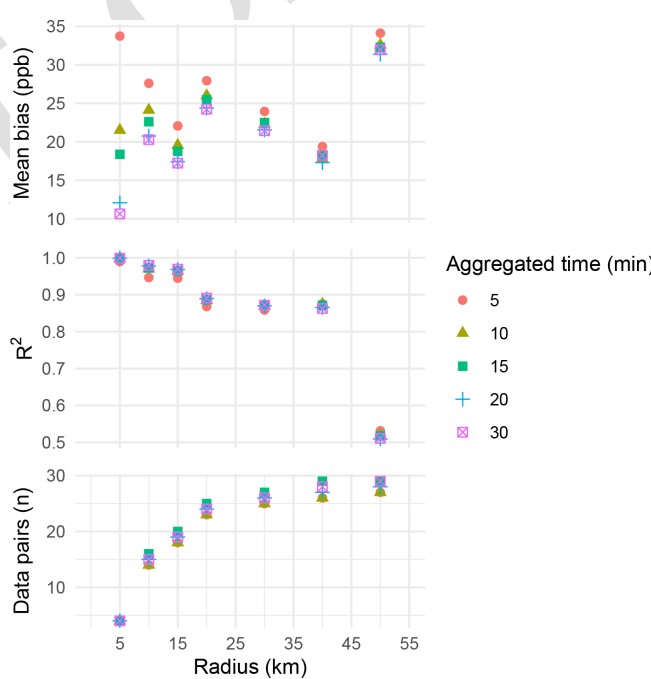

**Figure D1.** Results from the sensitivity analysis between the EM27/SUN and TROPOMI with a varying radius away from measurement site and varying aggregated times.

## Appendix E: Methane from wildfires

**Table E1.** Emissions of the 20 largest wildfires of 2020 in California. Estimates of $CO_2$ were obtained from CARB (2020). Emission ratios for Sierra Nevada fires (Creek, Castle, and North Complex) were derived from $EF_{CH_4}$ compiled in this study. CZU is for the San Mateo, Santa Cruz, and San Francisco counties. SCU is for the Santa Clara Unit. The rest of the $ER_{CH_4}$ are obtained from Xu et al. (2022), based on values from Prichard et al. (2020) and Xu et al. (2022).

| Fire name | General vegetation | Wildfire area burned (acres) | $CO_2$ (MMT) | $ER_{CH_4}$ (mol mol$^{-1}$) | $CH_4$ (Gg) |
|---|---|---|---|---|---|
| August Complex | Temperate evergreen | 1 032 700 | 27.7 | $0.0055 \pm 0.0044$ | $55.4 \pm 44.3$ |
| SCU Complex | Grasslands and savanna | 396 399 | 4.6 | $0.0043 \pm 0.0028$ | $7.2 \pm 4.7$ |
| Creek | Temperate evergreen | 379 882 | 13.8 | $0.0084 \pm 0.0022$ | $42.2 \pm 11$ |
| North Complex | Temperate evergreen | 318 777 | 10.9 | $0.0084 \pm 0.0022$ | $33.3 \pm 8.7$ |
| Hennessey | Shrublands | 305 352 | 3.5 | $0.0033 \pm 0.0021$ | $4.2 \pm 2.7$ |
| Castle | Temperate evergreen | 170 648 | 6.4 | $0.0084 \pm 0.0022$ | $19.5 \pm 5.1$ |
| Slater | Temperate evergreen | 157 430 | 6.7 | $0.0055 \pm 0.0044$ | $13.4 \pm 10.7$ |
| Red Salmon Complex | Temperate evergreen | 143 836 | 4.6 | $0.0055 \pm 0.0044$ | $9.2 \pm 7.4$ |
| Dolan | Shrublands | 124 527 | 2.1 | $0.0033 \pm 0.0021$ | $2.5 \pm 1.6$ |
| Bobcat | Shrublands | 115 998 | 2.5 | $0.0033 \pm 0.0021$ | $3.0 \pm 1.9$ |
| CZU Lightning Complex | Temperate evergreen | 86 553 | 5.4 | $0.0055 \pm 0.0044$ | $10.8 \pm 8.6$ |
| W-5 Cold Springs | Grasslands and savanna | 84 817 | 0.7 | $0.0043 \pm 0.0028$ | $1.1 \pm 0.7$ |
| Caldwell | Grasslands and savanna | 81 224 | 0.4 | $0.0043 \pm 0.0028$ | $0.6 \pm 0.4$ |
| Glass | Shrublands | 67 484 | 1.9 | $0.0033 \pm 0.0021$ | $2.3 \pm 1.4$ |
| Zogg | Shrublands | 56 338 | 0.7 | $0.0033 \pm 0.0021$ | $0.8 \pm 0.5$ |
| Wallbridge | Shrublands | 55 209 | 4.1 | $0.0033 \pm 0.0021$ | $4.9 \pm 3.1$ |
| River | Shrublands | 50 214 | 0.9 | $0.0033 \pm 0.0021$ | $1.1 \pm 0.7$ |
| Loyalton | Grasslands and savanna | 46 721 | 0.7 | $0.0043 \pm 0.0028$ | $1.1 \pm 0.7$ |
| Dome | Shrublands | 44 211 | 0.1 | $0.0033 \pm 0.0021$ | $0.1 \pm 0.1$ |
| Apple | Shrublands | 33 209 | 0.8 | $0.0033 \pm 0.0021$ | $1.0 \pm 0.6$ |
| Total | | | | | $213.7 \pm 49.8$ |

**Data availability.** The EM27/SUN retrievals used in this study are available at https://osf.io/ntzk8/ (last access: 15 June 2022 TS2). TROPOMI carbon monoxide and aerosol layer height products can be downloaded from https://s5phub.copernicus.eu (last access: 15 July 2022; ESA, 2022). We acknowledge the use of imagery from the NASA Worldview application (https://worldview.earthdata.nasa.gov/, last access: 15 July 2022; NASA, 2022a), which is part of the NASA Earth Observing System Data and Information System (EOSDIS). Version 3 AOD data are available from the AERONET website (https://aeronet.gsfc.nasa.gov, last access: 15 June 2022, NASA, 2022b). Fire radiative power data can be downloaded from https://firms.modaps.eosdis.nasa.gov/ (last access: 15 June 2022; NASA, 2022c). NOAA Physical Science Laboratory (PSL) wind data can be downloaded from ftp://ftp1.psl.noaa.gov/psd2/data/realtime/Radar915/ (last access: 15 June 2022; NOAA, 2022).

**Supplement.** The supplement related to this article is available online at: https://doi.org/10.5194/acp-23-1-2023-supplement.

**Author contributions.** IFV and SH contributed to the paper via conceptualization and data curation. IFV and HAP contributed to the data collection. IFV, SH, and AGM contributed via formal analysis. The publication was written by IFV, and all authors reviewed the paper and contributed to the discussion of the paper. FMH and MD contributed to funding acquisition.

**Competing interests.** At least one of the (co-)authors is a member of the editorial board of *Atmospheric Chemistry and Physics*. The peer-review process was guided by an independent editor, and the authors also have no other competing interests to declare.

**Special issue statement.** This article is part of the special issue "The role of fire in the Earth system: understanding interactions with the land, atmosphere, and society (ESD/ACP/BG/GMD/N-HESS inter-journal SI)". It is a result of the EGU General Assembly 2020, 4–8 May 2020.

**Acknowledgements.** Isis Frausto-Vicencio acknowledges the support from the National Science Foundation Graduate Research Fellowship Program. We thank the principal investigator Michael Garay and site manager Scott Scheller for their effort in establishing and maintaining the AERONET Fresno_2 site. We thank Nicole Jacobs for providing the code to apply the averaging kernel correction to the EM27/SUN $X_{CO}$ observations. We thank Jacob Hedelius for providing code through EGI to read micro-windows from EM27/SUN retrievals. Finally, the authors thank William Porter for the assistance and access to University of California, Riverside (UCR), Aldo cluster.

**Financial support.** This research has been supported by the Office of the President, University of California (grant no. LFR-18-548581).

**Review statement.** This paper was edited by Eduardo Landulfo and reviewed by two anonymous referees.

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

### Remarks from the language copy-editor

### Remarks from the typesetter