# Peer review of "Ground solar absorption observations of total column CO, CO2, CH4, and aerosol optical depth from California's Sequoia Lightning Complex Fire: Emission factors and modified combustion efficiency at regional scales"

_Atmospheric Chemistry and Physics, 2022_

## Author Comment (AC1)

**Response to Referee #1**

We would like to thank Reviewer 1 for carefully reading the manuscript and providing thorough comments.

**General comment 1:** "While the scientific asset of using such an instrument to study wildfire is explicit, discussion about the limitation of this study is clearly missing."

**Response to General Comment 1:** A paragraph describing the limitations of this study and instrument were added to the discussion: lines 455-477.

**General Comment 2:** "There are some repetitions in the paper and authors should reorganize some parts to make it more concise. For instance, some similar sentences are shown in different parts throughout the paper: lines 78-80 similar to lines 240-244, line 557-559 is quasi similar to lines 94-96, and 26-28."

**Response to General Comment 2:** We deleted lines 78-80 to make introduction more concise and is no longer similar to lines 240-244. We modified the lines 26-28, 94-96 and 557-559 to:

Lines 26-28: Our work demonstrates a novel application of the ground based EM27/SUN solar spectrometers in wildfire monitoring  by integrating regional scale measurements of trace gases and aerosols from smoke plumes.

Lines 557-559 (now lines 553-554): Overall, our analysis  contributes to the development of techniques for analyzing remotely sensed greenhouse gas measurements from wildfires.

**General Comment 3:** "The discussion section is interesting and well written. The introduction section should be more concise and emphasize the research state of art. More appropriate references are needed. I would recommend not using references in the conclusion section and focusing on summarizing the main results of this study. A paragraph in the conclusion is missing to highlight the limitations and the perspectives of this work."

**Response to General Comment 3:** Thank you for the suggestions. References were removed from the conclusion section. A paragraph was added to the discussion section (lines 455-477) to highlight the limitations of the EM27/SUN measurement technique.

"While advantages of this technique allow for understanding regional scale emissions, limitations exist with this method. The EM27/SUN solar column observations are limited to daytime hours as the instrument requires the sun as the light source. For this reason, we were not able to capture nighttime observations despite the continued release of smoke emissions and

growing concern of increasing nighttime wildfire activity in the continental United States (Freeborn et al., 2022). Additionally, optically thick smoke plumes obstruct the sunlight and prohibits continued measurements when the solar disk is not traceable by the instrument's solar tracker. Exposing the instrument's mirrors to harsh conditions such as ash depositing to ground observations on Sept. 12 decreases the instrument signal and may decrease the lifetime of mirrors. Although total column measurements are sensitive to larger scales than in situ stations, the FTIR is limited to the line of sight of the instrument and on occasions can miss the plume like we did on Sept. 13 and 14. Whereas aircraft observations have extensive spatial reach and more flexibility in locating and sampling plumes to obtain spatially rich information of the plume. However, when used in tandem with satellite observations our instrument collects continued temporal observations of a site of interest that a satellite does not, thus synchronous observations provide a better spatiotemporal understanding of the emission source. EM27/SUN instruments are also costly which can limit the number of instruments deployed. Unless instruments are secured properly as they have been done in long term network studies (Frey et al., 2019; Dietrich et al., 2021), measurements require personnel to set up and operate the instrument daily. The EFs, MCE, and their uncertainties fall within the range of expected values, thus lends confidence that this technique can be used for studying combustion phases of wildfires for other vegetation types. Despite the limitations of the EM27/SUN, we demonstrate the ability to gather new information of EF, MCE and AOD for understudied vegetation types and regions. Furthermore, the EM27/SUN observations can be used as a validation tool for orbiting satellites like TROPOMI, Orbiting Carbon Observatory-2 (OCO-2), OCO-3, and future satellites. The next generation weather forecasting, greenhouse gas, and air pollutant satellites such as Tropospheric Emissions: Monitoring of Pollution (TEMPO) will have more temporal frequency and greater spatial resolution allowing for continuous monitoring of burning activity and smoke emissions (Zoogman, 2017). This may allow remote sensing products to provide new insight into fuel properties of many types of vegetation in remote areas. It is will also be important to evaluate satellite-based observations with ground-based stations like the EM27/SUN as we did in this study."

**General Comment 4:** "Although authors are very thorough in the sensitivity tests, the main concerns are error estimations and background measurements. The error estimations are missing in this study. There are no estimation of the measurement uncertainties of Xgas and AOD which should be incorporated and propagated in the calculation of ER, EF and MCE. The slopes of the linear fits should reflect errors propagation and be mentioned with an error bar (+-)."

**Response to General Comment 4:** Thank you for this comment. We have incorporated uncertainties in the AOD estimation and included errors in the CO/AOD linear fit (lines 273-276), see Figure 3. AOD errors are described in the Appendix B, lines 575-575.

Error propagation was described for ER (lines 190-192) and EFs (lines 337-338). For the averaged MCE we reported the standard deviation but clarified that in line 339. We also show the slope error in Figure 5.

**General Comment 5:** "The background values are very important in the ER computation. How can you ensure that the 2nd percentile of the daily measured mixing ratios represent the background at this location? Why comparing the background SJV values to the very remote location of Mauna Loa? Could you find appropriate background values located at closer sites (Caltech, Mont Wilson, Dryden, other)? The ER values greatly depend on background concentrations and measurement precision. Error bars should be added to these estimates."

**Response to General Comment 5:** We agree with the reviewer that a more appropriate background location closer to the measurement site should be used, however during the 2020 fire season several wildfires occurred during the same period in Southern California. We explored using the southern California TCCON sites (Caltech and Armstrong/Dryden) as background sites, but $X_{CO}$ was elevated due to local wildfires in those areas and thus were not appropriate to use during this time. Mt. Wilson was also heavily fire impacted in September 2020, with flames reaching within 150 m of the observatory.

Leveraging the comparison between our ground-based instrument and TROPOMI, we compared the background measured from our site to the variation shown in the TROPOMI overpass. Figure S2 was added to the supplement where TROPOMI retrievals support the 2nd percentile background. Text and figure added to supplements:

**"Background estimation of EM27/SUN measurements**

The enhancement over background ($\Delta Xgas$) was calculated by subtracting the background ($Xgas, bkdg$) determined as the 2nd percentile of the daily measured mixing ratios ($Xgas$). Due to ongoing wildfires throughout the state, TCCON stations in Southern California (Caltech or NASA Armstrong/Dryden) were inappropriate as background sites. We used TROPOMI satellite retrievals of $X_{CO}$ on the Sept. 12 plume event as a case study to determine whether a 2nd percentile subtraction is appropriate.

TROPOMI satellite measurements can provide a better spatial understanding of heterogenous emissions during events like the wildfire plume. During the day on Sept. 12, the EM27/SUN measured a background of 220 ppb determined by the 2nd percentile (green line, Figure S2a). From TROPOMI observations, we can see that south of the plume was relatively "cleaner" while north of the plume $X_{CO}$ levels were higher (Figure S2b) due to emissions of multiple wildfires burning in the Sierra Nevada flowing southward. On this day, the EM27/SUN did not reach lower $X_{CO}$ levels as observed by TROPOMI and in Figure S2c we can see that the appropriate background for the EM27/SUN is determined by the instrument itself as the 2nd percentile.

[Figure]

a)

b)

c)

Figure S3. a) Timeseries of Sept. 12 plume event with green line representing the background (220 ppb) determined as the 2nd percentile of the daily measurements. TROPOMI satellite XCO retrievals of Sept 12 plume event with a) with location of EM27/SUN displayed with a magenta marker and a red line marking the 36° latitude line. b) Latitudinal TROPOMI $X_{CO}$ with red line showing the average 105 ppb $X_{CO}$ below the 36 latitudinal line and the green line displaying the background determined from the EM27/SUN 2nd percentile daily measurement.

**General Comment 6:** "Some figures could be improved for clarity purpose. Ex: the time series does not display well intraday variations."

**Response to General Comment 6:** Thank you for the suggestion. Based on similar comments made by reviewer #2, Figure 4 was added to Figure 2 and intraday variations are more visible.

**General Comment 7:** "The word "large scale" is recurrent in the manuscript and does not seem to be appropriated. Is regional more appropriated? The title should be modified since "large scales" is vague.

**Response to General Comment 7:** Thank you for the comment. We agree with the reviewer that the term is vague and have replaced the term "large scale" with "regional" throughout the text.

**Specific comments:**

1)  Please clarify the differences between Emission Ratios and Enhancement Ratios.

Response: We thank the reviewers for this comment and agree that clarification is necessary between emission ratio and enhancement ratios. We follow the definitions described in Yokelson et al., 2013 where a distinction is made between both concepts based on the conservation of the source ratio after the plume mixes with background air. We further elaborate below, and, in the manuscript in lines 365-369.

Enhancement ratios are also known as the normalized excess mixing ratios. Excess mixing ratios are calculated by subtracting the mixing ratio of a species from a source plume minus a mixing ratio of the same species in background air. To correct for dilution, excess mixing ratios are then normalized by a stable tracer, such as CO or $CO_2$. When an enhancement ratio does not change with dilution and mixing with background air, the enhancement ratio is equal to the emission ratio of a source.

In this study we measure stable compounds ($CH_4$, CO, and $CO_2$) that are not expected to chemically react in atmosphere during the duration of the local atmospheric transport. However, during this fire period many wildfires were burning throughout the state and smoke plume funneled into the SJV from other fires changing the background composition. Because of the SJV valley topography, air becomes stagnant, and pollution builds up, thus a true background was never reached for Sept. and Oct. 2020 measurements. We use Sept. 12 as a case study, where remote sensing instrument was directly underneath a thick smoke plume. We subtract the local background to isolate the plume in order to calculate an emission ratio for the fire.

2)  Abstract line 19: please define at "10km scales". Is it vertical or horizontal scale?

Response: Thank you for pointing this out. We have changed "10 km scales" to "10 km horizontal scales."

3)  Line 57: "fire conditions", please explain what conditions.

Response: Thank you for the comment. We have changed the vague term "fire conditions" to "wildfire combustion phases."

4)  Figure 3b: the error bars are the standard deviation. Errors on both TROPOMI and EM27/SUN measurements should be included in the linear fit.

Response: We used the York linear fit that incorporated errors in the x and y to calculate the slope and error. We have modified the figure (now Figure 4b) to show the slope with error.

5)  Section 3.3: authors state that FTIR and AERONET AOD are in agreement. What is the R value? How can you prove it?

Response: A scatterplot was added to the supplements (Figure S3) displaying a R value of hourly average comparisons.

6) Figure 4: reduce point size or find a solution to better display intraday variations.

Response: Figure 4 was merged with Figure 2 and focused on days Sept. 8 -15.

7) Figure 5: What are the measurement errors? Could you propagate the errors to obtain slopes values with all uncertainties?

Response: We used York linear regression to calculate the slope and error. The instrument errors for $X_{CO}$, $X_{CO2}$ and $X_{CH4}$ were used for this calculation and the Figure 5 was modified to show the slope with error. Text was added to clarify this in lines 190-193.

8) Figure 9: what is the error bar on the Top 20 CA wildfires emissions?

Response: The error shown on the Top 20 CA wildfire $CH_4$ emissions (now Figure 8) was calculated by propagating the $ER_{CH4}$ error from Table E1 into each individual wildfire CH4 estimate and added in quadrature to obtain a total error. More detail was added in the text in lines 424 - 426.

9) Please verify the order of the references in the parenthesis throughout the manuscript. It should follow the ACP journal recommendations: https://www.atmospheric-chemistry-and-physics.net/submission.html#references (ex: line 72; lines 110-111; …)

Response: We have verified the order of references throughout the manuscript.

10) Title of section 3.5 should be more specific. SJV GHG's sources are only dairy farms?

Response: Section 3.5 was split into two sections: "Enhancement ratios of livestock and wildfire emissions" and "Total methane emissions from wildfires in California."

**Technical Corrections**

1) Line 34-35: Rephrase this sentence and define particulate matter 2.5.

Response: This sentence was rephrased and particulate matter 2.5 was defined.

2) Line 56: Is reference "CARB 2018" appropriate?

Response: We have updated the reference to an appropriate reference: Lasslop et al., 2019.

3) Line 51: Is IPCC 2014 correct? Can you refer to a more recent report?

Response: Thank you for your comment. We have updated "IPCC 2014" to "IPCC 2021".

4) Line 65-66, 69, 70, 71 and more: add references.

Response: More references were added.

Lines 65-66:  Schneising et al., 2020, Whitburn et al., 2015, Adams et al., 2019, Griffin et al., 2021, and Jin et al., 2021

Line 69 (now line 72): Chen et al., 2016 and Heerah et al., 2021

Line 70 (now line 73): Frey et al., 2019, Vogel et al., 2019, Alberti et al., 2022a, and Alberti et al., 2022b

Line 71 (now line 74-75): Bader et al., 2017 and De Mazière et al., 2018

5) Figure 5, 8: Change dots color or size to clearly display all the points.

Response: Points on Figures 5 and 8 (now Figure 3 and 7) were reduced in size to clearly display all points.

6) Figure 3b, 6, D1: R2 should be R2

Response: Thank you for your suggestion. The "R2" in Figures 3b, 6, D1 was switched to $R^2$.

---

## Author Comment (AC2)

**Response to Referee #2**

We would like to thank Reviewer 2 for carefully reading the manuscript and providing thorough comments.

**Specific Comments**

1)  Line 25: When are 2020 $CH_4$ emissions expected to be available? (Consider updating to 2020, if the estimates are available before publication.)

Response: Thank you for this comment. The 2020 $CH_4$ emissions were recently published, and estimates were updated to 2020.

2)  Lines 26, 94, 557: "a novel application" – this statement is vague and, should specify what is novel about the application compared with previous studies.

Response: We have specified the novelty of this method in line 26: "Our work demonstrates a novel application of the ground-based EM27/SUN solar spectrometers in wildfire monitoring by integrating regional scale measurements of trace gases and aerosols from smoke plumes."

Line 94: Deleted phrase.

Line 557 (now line 553): "Overall, our analysis contributes to the development of techniques for analyzing remotely sensed greenhouse gas and aerosol measurements from wildfires."

3)  Line 62-64: "observations… focus on aerosol burden from smoke plumes with limited attention to trace gases…" This isn't entirely true - the paper should indicate that there have been a number of studies that have looked at trace gases emitted from fires using satellite data, including ratios of species and estimation of emissions for CO, NOx, NH3 (see for example, Griffin et al., AMT 2021; Adams et al. ACP 2019; Whitburn et al., Atmos. Env., 2015 and references therein).

Response: The text was modified to include trace gas studies and included the recommended references:

"While several space-based instruments can retrieve and derive emissions of important trace gases globally, observations of trace gases are limited by spatiotemporal coverage and aerosol burden from smoke plumes (Schneising et al., 2020). Recent satellite studies have focused on trace gas emissions and ratios for $CH_4$, CO, nitrogen oxides ($NO_x$) and ammonia ($NH_3$) (Whitburn et al., 2015; Adams et al., 2019; Griffin et al., 2021; Jin et al., 2021), but few focus on the integration of trace gases and aerosols."

4)  Line 65 : ".. present a new technique…" Is ground-based FTIR is a particularly new technique?

Response: Thank you for the comment. We have changed "present a new technique" to "present an alternate technique." The ground-based FTIR is not a new technique and has been used

previously in other studies to estimate emissions of wildfires, however it is the first time that emission factors with respect to $CO_2$ have been calculated with this ground-based remote sensing technique.

5) Line 91: "… satellite greenhouse gas observations…" perhaps replace with "… observations of CO…"

Response: This has been changed to "observations of CO."

6) Line 97: I found this section a little bit difficult to follow. Recommend giving an overview of the various instruments involved (e.g., describing of Fig 1 to a high level) and the description of the fire to the top of the section (before Sect. 2.1), and then using the subsections to give the technical details. I would recommend that you describe/name the fires here and then use consistent naming throughout the document. (For example, the name "Shotgun" fire is used in some places – is this the same as the "North Complex" fire in Table E1?)

Response: This section was modified per reviewer's suggestion of describing Fig. 1 prior to Section 2.1 followed by subsections describing technical details of instruments used.

7) Line 150: Should panels b, c of the figure be referenced/described here?

Response: Apologies if we misunderstood but panels a, b, and c are described in the figure.

8) Line 163: "… for a novel evaluation…" – is this evaluation novel? Has TROPOMI CO not been evaluated under smoky conditions before?

Response: We have modified this sentence to add more context of the importance and novelty of our study in lines 166-168.

"There is growing intertest in using the TROPOMI XCO product for understanding global wildfire fluxes, however few studies focus on evaluating those observations (e.g., Jacobs 2021 and Rowe et al., 2021). We measured a range of $X_{CO}$ levels of mixed smoke plumes and were able to isolate a concentrated smoke plume from a nearby fire. This allowed for a ground-based evaluation of the TROPOMI sensor under various wildfire conditions, including high $X_{CO}$ and aerosol loading in the atmosphere."

9) Line 180: Should Sect 2.4 be moved into an appendix, since it mostly references methods described elsewhere?

Response: Thank you for the suggestion. We have moved Section 2.4 to Appendix B.

10) Line 295: For the comparisons against TROPOMI should uncertainties be included in the fits, etc?

Response: We fit the comparison between the two instruments with a York linear regression that considers uncertainties in both x and y. We added the uncertainty in the slope of the linear fit on Figure 4b.

11) Line 307 (and similar statements lines 500, 546): "These results suggest an overestimation of 9.7% X_CO from TROPOMI observations of wildfires." Is there evidence that the difference is due to overestimation from TROPOMI? Or could this be due to differences in sampling or biases in the EM27/SUN data? Is there an uncertainty attached to the 9.7%? Also, what is the reported uncertainty in TROPOMI measurements and in the EM27/SUN measurements? Is 9.7% within the range of uncertainties? Is this bias consistent with previous studies?

Response: According to Sha et al., 2021, the systematic difference $X_{CO}$ between TCCON and TROPOMI observations is on average $9.22 \pm 3.45\%$. Our estimate of $9.7 \pm 1.3\%$ is very close to the systematic difference reported by Sha et al., 2021, however based on our sensitivity study biases may exist based on sampling conditions in a spatially and temporally heterogenous source. We included this information in lines: 318-322.

12) Line 332 (Figure 4): Can figure 4 be added to figure 2 as a panel? Could be helpful to see all the timeseries together and would reduce the number of figures needed.

Response: We have added Figure 4 as panel e in Figure 2.

13) Line 339: "McMillan et al. (2008) found values…" should the word "values" be replaced with "slopes"

Response: The word "values" were replaced with "slopes."

14) Line 340: "… 40 to 74…" should units be provided for this?

Response: Units ppb $X_{CO}$/AOD were added to the slopes.

15) Lines 339-343: I find the discussion in these sentences a bit confusing and could be organized a bit more clearly. For example, are the values from McMillan et al of 44-74 the same as the values that are for clean region described further down the paragraph?

Response: Thank you for the comment. The discussion in this section was rewritten (lines 271-288) for clarity and a table of summary statistics (Table 1) was added to provide more detail.

16) Line 348 (Figure 5 caption): There is no "teal line" on the figure – does this mean the "teal markers"?

Response: We changed "teal line" to "teal markers."

17) Line 356: "… a steady MCE as X_CO, X_CH4, and AOD increased, indicating influence of smoldering combustion" Please elaborate – why does this indicate smoldering?

Response: Figure 5, panel a-e was removed as suggested below by reviewer, thus this sentence was removed as it was referring to the timeseries in panels a-e.

18) Lines 365-371: The discussion on EFs should be merged with the discussion on lines 458-464. I also find parts of the discussion a bit hard to follow. Which studies were included in the table and why? Which values are most relevant for comparison against the present study? How does Lueker et al., 2001 compare to the results in this study?

Response: Lines 365-371 were merged with lines 458-464.

The discussion was rewritten to clarify the main points. We also included context as to why the studies were chosen to be included in Table 2 and their relevancy for comparison against this study. Our results were consistent with $CH_4$ emissions found in Lueker et al., 2001 (line 446).

19) Line 372 (Table 1 caption): Should mention the present study in the caption, e.g., "Summary of past airborne studies, and the present study…"

Response: Added.

20) Line 381 (Figure 6): Do panels a-e add value to the paper? Is the same/similar information captured in the broader timeseries in Fig 2?

Response: We have removed panels a-e. The data shown on panels a-e is the same as in Fig.2 and does not add value to the paper.

21) Line 388 (Figure 7): If including this figure in the paper, should describe its relevance in the text.

Response: We have described the relevance of Figure 7 (now Figure 6) in the text in lines 332-336.

22) Line 389 (Sect. 3.5 title): This section title is vague. Perhaps split Sect. 3.5 into two sections (one section about ratios for livestock vs wildfire emissions and another section about estimating total methane emissions from wildfires in California?)

Response: Section 3.5 was split into two sections: "Enhancement ratios of livestock and wildfire emissions" and "Total methane emissions from wildfires in California."

23) Lines 401-411: I found the discussion of the different ratios difficult to follow, and could use a rewrite for clarity. For example on line 407, "Similar ratios… were found in Hanford…" – I assume this means ratios similar to the non-wildfire ratios?

Response: This was paragraph rewritten and shorten for clarity (lines 376-389).

24) Line 414-415: "… dairy farm operations are the dominant source of CH4 during fire and non-fire periods." This seems to contradict the next sentence, which says that during the strong smoke influence period, CH4 from the smoke is dominant. Clarify.

Response: Thank you for the suggestion. We clarified these statements by specifying that they compare between fire and non-fire days.

25) Line 416: "The immense scale…" recommend starting a new section (or at least a new paragraph) here. Lines 430-435 and 635-640: Please provide a bit more information about how Table E1 was filled in. Why is the ER from the study in table E1 (0.0084) different from the ER given in line 362 (0.0073)? How were the ERs derived from the EFs in the literature? How were uncertainties propagated? Which values in Table E1 correspond to Pritchard and which correspond to Xu? Were there any cases where both Xu and Pritchard had different ER values for the same vegetation, and if so how did you choose which to use? (Also, check that everything is consistent between the text and the appendix: Line 434 references Xu 2020, but line 634 table caption references Prichard 2020 and Xu 2022).

Response: A new section was started after "The immense scale."

We agree that the wording in the preprint manuscript was confusing for how Table E1 was filled out. For non-Sierra Nevada wildfires, literature values were used for three different vegetation types. The literature EF values for general vegetation types were obtained from mean values in Xu et al., 2022 that were based on EF's found in Prichard et al., 2020. We calculated the standard deviation for the mean EF values from the Prichard et al., 2020 and propagated this as the error into the calculated ER values. We have clarified this in the main text and in the appendix.

In our approach to calculate an ER for the Sierra Nevada wildfires, we took an average of the EF from Sierra Nevada studies in Table 1 (EF$_{CH4\_avg}$ = 5.6 ± 1.5 g kg$^{-1}$). From the EF$_{CH4\_avg}$ we derived an ER$_{CH4\_avg}$ = 0.0084 ± 0.0022 that was calculated from Equation 2 by solving for ER with C$_T$ equal to 1. The ER$_{SQF}$ that we calculated from Sept 12 plume (0.0073) was part of the averaged ER was not used directly as part of Table E1 calculations, rather it was included in the ER$_{CH4\_avg}$. We believe using ER$_{CH4\_avg}$ would better represent temperate vegetation wildfires in the Sierra Nevada. We have added those values to the paragraph to clarify our process for calculations in lines 418-419.

Thank you for catching that mistake, line 434 (now line 415) should have been Xu et al., 2022.

26) Lines 454-457: Repetitive – delete summary of the work and save this for the conclusion/abstract?

Response: The summary of work was removed from the discussion.

27) Lines 471-487: This discussion feels a biased toward FTIR measurements. What are the drawbacks to FTIR measurements compared to the other methods? Is there information that can be provided by aircraft that can't be provided by FTIR? Do all of these methods have similar uncertainties in, e.g., emission factors? Are the FTIRs more difficult to operate than say continuous ground-based in-situ analyzers? Is there potential for satellite (especially the next generation of satellites), modelling, or other methods to add to knowledge on fire emission factors as well?

Response: Thank you for this comment. We have rewritten this paragraph to include drawbacks of this measurement technique in lines 455-477:

"While advantages of this technique allow for understanding regional scale emissions, limitations exist with this method. The EM27/SUN solar column observations are limited to daytime hours as the instrument requires the sun as the light source. For this reason, we were not able to capture nighttime observations despite the continued release of smoke emissions and growing concern of increasing nighttime wildfire activity in the continental United States (Freeborn et al., 2022). Additionally, optically thick smoke plumes obstruct the sunlight and prohibits continued measurements when the solar disk is not traceable by the instrument's solar tracker. Exposing the instrument's mirrors to harsh conditions such as ash depositing to ground observations on Sept. 12 decreases the instrument signal and may decrease the lifetime of mirrors. Although total column measurements are sensitive to larger scales than in situ stations, the FTIR is limited to the line of sight of the instrument and on occasions can miss the plume like we did on Sept. 13 and 14. Whereas aircraft observations have extensive spatial reach and more flexibility in locating and sampling plumes to obtain spatially rich information of the plume. However, when used in tandem with satellite observations our instrument collects continued temporal observations of a site of interest that a satellite does not, thus synchronous observations provide a better spatiotemporal understanding of the emission source. EM27/SUN instruments are also costly which can limit the number of instruments deployed. Unless instruments are secured properly as they have been done in long term network studies (Frey et al., 2019; Dietrich et al., 2021), measurements require personnel to set up and operate the instrument daily. The EFs, MCE, and their uncertainties fall within the range of expected values, thus lends confidence that this technique can be used for studying combustion phases of wildfires for other vegetation types. Despite the limitations of the EM27/SUN, we demonstrate the ability to gather new information of EF, MCE and AOD for understudied vegetation types and regions. Furthermore, the EM27/SUN observations can be used as a validation tool for orbiting satellites like TROPOMI, Orbiting Carbon Observatory-2 (OCO-2), OCO-3, and future satellites. The next generation weather forecasting, greenhouse gas, and air pollutant satellites such as Tropospheric Emissions: Monitoring of Pollution (TEMPO) will have more temporal frequency and greater spatial resolution allowing for continuous monitoring of burning activity and smoke emissions (Zoogman, 2017). This may allow remote sensing products to provide new insight into fuel properties of many types of vegetation in remote areas. It is will also be important to evaluate satellite-based observations with ground-based stations like the EM27/SUN as we did in this study."

28) Line 491: "with great resolution" – subjective (remove or replace this).

Response: The phrase "with great resolution" was change to "high spatial resolution."

29) Line 520-529: Can more context be provided on the discussion of wildfire emissions of CH4? Are there other estimates of CH4 emissions from wildfires in the California or is a data gap? When discussing possible climate feedbacks, is it expected that a lot of CH4 is emitted from fires globally? Are CH4 emissions from fires considered, for example, in IPCC reports?

Response: Thank you for this comment. We have added more context of $CH_4$ wildfire emissions, discussed data gaps, global $CH_4$ emissions and global inventories including the IPCC report in lines 511 – 527.

**Technical Corrections**

1) Line 490: "12. Smoke event" (delete the period after 12)

Thank you for catching that. The period was deleted.